# OVERCLOCKING ELECTROSTATIC GENERATIVE MODELS

## ABSTRACT

Electrostatic generative models such as PFGM++ have recently emerged as a powerful framework, achieving state-of-the-art performance in image synthesis. PFGM++ operates in an extended data space with auxiliary dimensionality $D$, recovering the diffusion model framework as $D \to \infty$, while yielding superior empirical results for finite $D$. Like diffusion models, PFGM++ relies on expensive ODE simulations to generate samples, making it computationally costly. To address this, we propose Inverse Poisson Flow Matching (IPFM), a novel distillation framework that accelerates electrostatic generative models across all values of $D$. Our IPFM reformulates distillation as an inverse problem: learning a generator whose induced electrostatic field matches that of the teacher. We derive a tractable training objective for this problem and show that, as $D \to \infty$, our IPFM closely recovers Score Identity Distillation (SiD), a recent method for distilling diffusion models. Empirically, our IPFM produces distilled generators that achieve near-teacher or even superior sample quality using only a few function evaluations. Moreover, we observe that distillation converges faster for finite $D$ than in the $D \to \infty$ (diffusion) limit, which is consistent with prior findings that finite-$D$ PFGM++ models exhibit more favorable optimization and sampling properties.

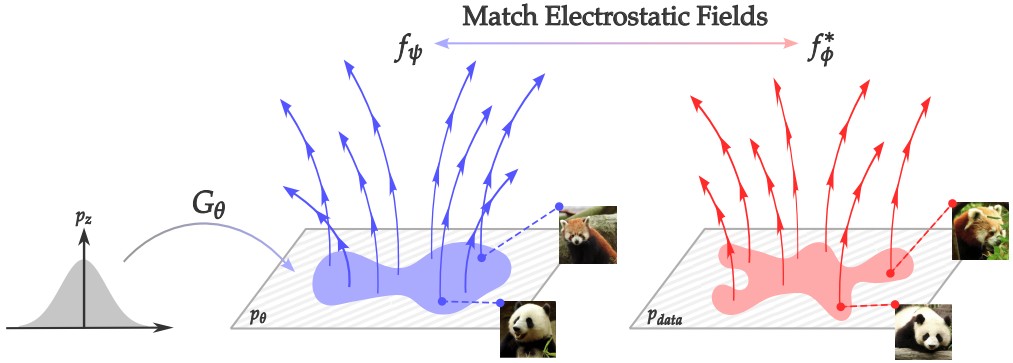

Figure 1: **Our IPFM overview.** We seek a generator $G_\theta$ whose distribution $p_\theta(\mathbf{y})$ induces an electrostatic field $f_\psi$ that matches the field $f_\phi^*$ of the teacher model trained on real data $p_{data}(\mathbf{y})$.

## 1 INTRODUCTION

Diffusion models (Sohl-Dickstein et al., 2015; Ho et al., 2020; Song et al., 2020) have emerged as a leading paradigm in generative modeling, achieving state-of-the-art performance in synthesizing high-quality samples across various domains. Inspired by non-equilibrium thermodynamics, these models gradually perturb data into a simple noise distribution through a forward process and learn to reverse this process, enabling high-fidelity generation.

A novel class of generative models inspired by *electrostatics* has recently emerged. The core idea is to treat data points as electric charges and evolve the distribution under a Coulomb-like field. The

Poisson Flow Generative Model (Xu et al., 2022, PFGM) leverages this analogy for noise-to-data generation: by placing the data distribution on a hyperplane in an augmented space, the resulting electrostatic field defines a dynamics that transform a uniform (on a hemisphere) distribution back into the data. A subsequent approach, Electrostatic Field Matching (Kolesov et al., 2025), extends this to data-to-data translation. It models two distributions as opposing charges whose interaction, governed by the superposition principle, generates an electrostatic field.

PFGM++ (Xu et al., 2023) proposes a unified framework for noise-to-data generation based on electrostatic principles that encompasses both PFGM and diffusion models as special cases. Specifically, the authors introduce an expanded space whose dimensionality depends on a parameter $D$, showing that PFGM++ reduces to PFGM when $D = 1$ and to diffusion models in the limit as $D \to \infty$. The authors find that models with smaller values of $D$ are more robust to numerical integration errors during sampling, while those with larger $D$ are easier to train. They further demonstrate that a *sweet spot* exists in the choice of $D$ that balances these properties, leading to improved generation quality. As a result, its competitive performance and emergent properties make the electrostatic-based modeling framework a promising and compelling direction for research.

However, like diffusion models, electrostatic models rely on expensive ODE simulations, making them computationally costly (e.g., 35 and 79 neural network evaluations for CIFAR-10 (Krizhevsky et al., 2009) and FFHQ 64x64 (Karras et al., 2019), respectively). This motivates developing techniques that reduce sampling cost while preserving generation quality.

To address this challenge, we introduce our **I**nverse **P**oisson **F**low **M**atching (IPFM), a novel distillation framework designed to accelerate PFGM++ models across all values of the auxiliary dimension $D$. An overview of the method is presented in Figure 1.

**Our main contributions** are as follows:

- **Theory & Methodology.** We formulate the distillation of PFGM++ as an inverse Poisson flow matching problem and derive a tractable objective applicable to all values of the auxiliary dimension $D$. We reveal a close connection between our IPFM in diffusion ($D \to \infty$) setting and Score Identity Distillation (Zhou et al., 2024, SiD), a recent method for distilling diffusion models.

- **Practical.** We demonstrate empirically that our IPFM enables distilled generators to match or surpass teacher sample quality in just a few steps. SiD-inspired regularization further improves these results. We also observe faster convergence for finite $D$ compared to the diffusion limit ($D \to \infty$), highlighting the favorable generation properties of finite-$D$ settings.

For completeness, we note that general ODE-based, e.g., consistency-like (Song & Dhariwal, 2023; Kim et al., 2023), distillation methods are also applicable to PFGM++. Recent work Poisson Flow Consistency Models (Hein et al., 2025, PFCM) adapts Consistency Distillation (Song et al., 2023) to PFGM++ for low-dose CT image denoising. However, for the diffusion ($D \to \infty$) case, it is known that such distillations are outperformed by SiD (Zhou et al., 2024, Section 5), especially when no auxiliary GAN-loss is applied. Since our IPFM reduces to SiD in the diffusion ($D \to \infty$) limit, we expect our IPFM to similarly surpass general ODE-based techniques in finite-$D$ settings. We confirm this intuition with a sanity-check experiment, which shows that our IPFM indeed outperforms a strong consistency-based baseline in the finite-$D$ case (see Appendix D.1).

## 2 BACKGROUND

In this section, we first recall the fundamental concepts of high-dimensional electrostatics (§2.1). Then we describe the state-of-the-art electrostatic approach PFGM++ (§2.2) for noise-to-data generation, which serves as a teacher model for our distillation framework. Finally, we cover PFGM++'s relationship to diffusion models (§2.3) and their denoising reparameterizations (§2.4).

### 2.1 HIGH-DIMENSIONAL ELECTROSTATICS

This section covers the fundamental principles of electrostatics necessary for understanding electrostatic-based generative models. For further details on electrostatics and its high-dimensional generalizations, see (Caruso et al., 2023; Ehrenfest, 1917; Gurevich & Mostepanenko, 1971).

**Electrostatic Field.** Consider a point charge $\mathbf{y} \in \mathbb{R}^N$ with charge $q$. According to Coulomb's law, this charge generates an electrostatic field given by:

$$\mathbf{E}(\mathbf{x}) = \frac{q}{S_{N-1}} \frac{\mathbf{x} - \mathbf{y}}{\|\mathbf{x} - \mathbf{y}\|_2^N}, \tag{1}$$

where $S_{N-1}$ is the surface area of the $(N-1)$-dimensional unit sphere. For a continuous charge distribution $q(\mathbf{y})$, the total field is obtained by integrating according to the superposition principle:

$$\mathbf{E}(\mathbf{x}) = \frac{1}{S_{N-1}} \int \frac{\mathbf{x} - \mathbf{y}}{\|\mathbf{x} - \mathbf{y}\|_2^N} q(\mathbf{y}) d\mathbf{y}. \tag{2}$$

**Electrostatic Field Lines.** An electrostatic field line is a curve $\mathbf{x}(t) \in \mathbb{R}^N$, parameterized by $t \in [a, b] \subset \mathbb{R}$, whose tangent vector is aligned with the electric field at each point. Such a curve satisfies the ordinary differential equation (ODE)

$$\frac{d\mathbf{x}(t)}{dt} = \mathbf{E}(\mathbf{x}(t)). \tag{3}$$

These field lines represent the trajectories that a positive test charge follows under the influence of the electrostatic field.

## 2.2 POISSON FLOW GENERATIVE MODEL (PFGM++)

In electrostatic generative modeling, the data distribution $p(\mathbf{y})$ is treated as a positive charge distribution, following the principles in §2.1. In this section, we review the state-of-the-art PFGM++ framework (Xu et al., 2023), which builds upon this foundation.

**Augmented space.** Given an $N$-dimensional data distribution $p(\mathbf{y})$, PFGM++ embeds it into an augmented space $\mathbb{R}^{N+D}$ by introducing $D$ auxiliary dimensions. Each data point $\mathbf{y} \in \mathbb{R}^N$ is placed on the hyperplane $\mathbf{z} = \mathbf{0}$, resulting in an extended point $\tilde{\mathbf{y}} = (\mathbf{y}, \mathbf{0}) \in \mathbb{R}^{N+D}$, where $D \in \mathbb{Z}^+$. The electrostatic field at a point $\tilde{\mathbf{x}} = (\mathbf{x}, \mathbf{z})$ is then given by

$$\mathbf{E}(\tilde{\mathbf{x}}) = \frac{1}{S_{N+D-1}} \int \frac{\tilde{\mathbf{x}} - \tilde{\mathbf{y}}}{\|\tilde{\mathbf{x}} - \tilde{\mathbf{y}}\|_2^{N+D}} p(\mathbf{y}) \, d\mathbf{y}. \tag{4}$$

Rather than tracking the full vector $\mathbf{z}$, the dynamics induced by the electric field (4) are captured by the radial component $r(\tilde{\mathbf{x}}) = \|\mathbf{z}\|_2$. This is motivated by the evolution of each auxiliary coordinate as $dz_i = \mathbf{E}(\tilde{\mathbf{x}})_{z_i} dt$, and similarly for the variable $r$ (Xu et al., 2023, Section 3.1). Thus, considering the augmented space consisting of points $(\mathbf{x}, r)$ enables a unified treatment of different values of $D$. Note that, as a result, PFGM++ works with objects $\tilde{\mathbf{x}} = (\mathbf{x}, r)$ and $\mathbf{E}(\tilde{\mathbf{x}})$ that lie in $\mathbb{R}^{N+1}$, while $D$ influences the geometry of the field via the exponent of the distance in (4).

**Sampling.** In PFGM++, the authors show that given the electrostatic field $\mathbf{E}(\tilde{\mathbf{x}})$, the data distribution can be reconstructed by following the dynamics:

$$\frac{d\mathbf{x}}{dr} = \frac{\mathbf{E}(\tilde{\mathbf{x}})_{\mathbf{x}}}{\mathbf{E}(\tilde{\mathbf{x}})_r}, \tag{5}$$

starting from the prior distribution

$$p_{r_{\max}}(\mathbf{x}) = \frac{r_{\max}^D}{(\|\mathbf{x}\|_2^2 + r_{\max}^2)^{\frac{N+D}{2}}}, \tag{6}$$

in the limit as $r_{\max} \to \infty$, or for a sufficiently large $r_{\max}$ in practice. Note that this ODE (5) is equivalent to moving data along the electrostatic field lines (3) but with adjusted speed:

$$d\tilde{\mathbf{x}} = d(\mathbf{x}, r) = \left( \frac{d\mathbf{x}}{dt} \frac{dt}{dr} dr, dr \right) = \left( \mathbf{E}(\tilde{\mathbf{x}})_{\mathbf{x}} \mathbf{E}(\tilde{\mathbf{x}})_r^{-1}, 1 \right) dr. \tag{7}$$

**Electrostatic Field Estimation.** From (5), one can see that only the normalized field direction is needed for sampling. To learn it, PFGM minimizes the following objective:

$$\mathbb{E}_{p_{\text{train}}(\tilde{\mathbf{x}})} \left\| f_\phi(\tilde{\mathbf{x}}) - \frac{\hat{\mathbf{E}}(\tilde{\mathbf{x}})}{\|\hat{\mathbf{E}}(\tilde{\mathbf{x}})\|_2} \right\|_2^2,$$

where $f_\phi : \mathbb{R}^{N+1} \to \mathbb{R}^{N+1}$, $p_{\text{train}}(\tilde{\mathbf{x}})$ is a heuristic distribution covering the volume between $r = 0$ and $r = r_{\max}$, and $\hat{\mathbf{E}}(\tilde{\mathbf{x}})$ is a sample-based estimator of the electrostatic field (4) at a point $\tilde{\mathbf{x}}$. However, this objective requires prohibitively large batches (effectively the full dataset) to yield an unbiased estimate of the target, i.e., the normalized field at a given point (Xu et al., 2023, Section 3).

To remedy these issues, PFGM++ proposes a perturbation-based objective, which is analogous to the objective used in diffusion models:

$$\mathbb{E}_{p(r)}\mathbb{E}_{p(\mathbf{y})}\mathbb{E}_{p_r(\mathbf{x}|\mathbf{y})} \|f_\phi(\tilde{\mathbf{x}}) - (\tilde{\mathbf{x}} - \tilde{\mathbf{y}})\|_2^2, \tag{8}$$

where $f_\phi : \mathbb{R}^{N+1} \to \mathbb{R}^{N+1}$, $r \in (0, r_{\max})$, $p(r)$ is the training distribution over $r$, and $p_r(\mathbf{x}_r|\mathbf{y})$ is a perturbation kernel of the form

$$p_r(\mathbf{x}_r|\mathbf{y}) \propto \frac{1}{\left(\|\mathbf{x}_r - \mathbf{y}\|_2^2 + r^2\right)^{\frac{N+D}{2}}}, \tag{9}$$

with $\tilde{\mathbf{y}} = (\mathbf{y}, 0)$ and $\tilde{\mathbf{x}} = (\mathbf{x}_r, r)$ denoting the clean and perturbed data points, respectively. A practical procedure for sampling from the kernel is presented in (Xu et al., 2023, Appendix B). The minimizer of this objective can be shown to be proportional to the electrostatic field (4):

$$f_\phi^*(\tilde{\mathbf{x}}) = \frac{S_{N+D-1}}{p_r(\mathbf{x})} \mathbf{E}(\tilde{\mathbf{x}}), \tag{10}$$

where $p_r(\mathbf{x}_r) = \int p_r(\mathbf{x}_r|\mathbf{y})p(\mathbf{y})\, d\mathbf{y}$ is the marginal distribution induced by the perturbation kernel $p_r(\mathbf{x}_r|\mathbf{y})$. This minimizer can be used to obtain the desired dynamics (5):

$$\frac{d\mathbf{x}}{dr} = \frac{f_\phi^*(\tilde{\mathbf{x}})_\mathbf{x}}{f_\phi^*(\tilde{\mathbf{x}})_r} = \frac{\left(\frac{S_{N+D-1}}{p_r(\mathbf{x})} \mathbf{E}(\tilde{\mathbf{x}})\right)_\mathbf{x}}{\left(\frac{S_{N+D-1}}{p_r(\mathbf{x})} \mathbf{E}(\tilde{\mathbf{x}})\right)_r} = \frac{\frac{S_{N+D-1}}{p_r(\mathbf{x})} \mathbf{E}(\tilde{\mathbf{x}})_\mathbf{x}}{\frac{S_{N+D-1}}{p_r(\mathbf{x})} \mathbf{E}(\tilde{\mathbf{x}})_r} = \frac{\mathbf{E}(\tilde{\mathbf{x}})_\mathbf{x}}{\mathbf{E}(\tilde{\mathbf{x}})_r}. \tag{11}$$

Note that $\tilde{\mathbf{x}} - \tilde{\mathbf{y}}$ in objective (8) equals $(\mathbf{x}_r - \mathbf{y}, r)$. If we divide it by $r/\sqrt{D}$, the last component becomes constant ($\sqrt{D}$) and can be omitted. Thus, the objective can be reduced to:

$$\mathbb{E}_{p(r)}\mathbb{E}_{p(\mathbf{y})}\mathbb{E}_{p_r(\mathbf{x}_r|\mathbf{y})} \left\| f_\phi(\tilde{\mathbf{x}}) - \frac{\mathbf{x}_r - \mathbf{y}}{r/\sqrt{D}} \right\|_2^2, \tag{12}$$

where $f_\phi : \mathbb{R}^{N+1} \to \mathbb{R}^N$. In this case, data generation follows the dynamics:

$$\frac{d\mathbf{x}}{dr} = \frac{f_\phi^*(\tilde{\mathbf{x}})}{\sqrt{D}}. \tag{13}$$

## 2.3 RELATION TO DIFFUSION MODELS

PFGM++ encompasses classic diffusion models (Xu et al., 2023, Section 4) within its framework. Specifically, in the limit $D \to \infty$ with the reparameterization $\sigma = r/\sqrt{D}$:

1. The PFGM++ ODE (5) converges to the diffusion ODE (Karras et al., 2022):

$$\frac{d\mathbf{x}}{dt} = -\dot{\sigma}(t)\sigma(t)\nabla_\mathbf{x} \log p_{\sigma(t)}(\mathbf{x}); \tag{14}$$

2. The PFGM++ perturbation kernel (9) converges to the Gaussian kernel used in diffusion models:

$$p_\sigma(\mathbf{x}_\sigma|\mathbf{y}) \propto \exp\left(-\frac{\|\mathbf{x}_\sigma - \mathbf{y}\|_2^2}{2\sigma^2}\right); \tag{15}$$

3. The PFGM++ objective minimizer (12) converges to the diffusion objective minimizer:

$$\mathbb{E}_{p(\sigma)}\mathbb{E}_{p(\mathbf{y})}\mathbb{E}_{p_\sigma(\mathbf{x}_\sigma|\mathbf{y})}\left\|f_\phi(\mathbf{x}_\sigma,\sigma) - \frac{\mathbf{x}_\sigma - \mathbf{y}}{\sigma}\right\|_2^2. \tag{16}$$

Moreover, the authors showed that the reparameterization $r = \sigma\sqrt{D}$ allows hyperparameters tuned for diffusion models ($D \to \infty$) to be transferred directly to finite $D$ settings.

## 2.4 Denoising Model Reparameterization

In practice, PFGM++ is trained using a *denoising network* $\hat{\mathbf{y}}_\phi(\mathbf{x}_r, r)$ that predicts the clean data $\mathbf{y}$ from a perturbed sample $\mathbf{x}_r$. The normalized field estimator is then computed as

$$f_\phi(\mathbf{x}_r, r) = \frac{\mathbf{x}_r - \hat{\mathbf{y}}_\phi(\mathbf{x}_r, r)}{r/\sqrt{D}}.$$

Substituting this into the PFGM++ objective (12) yields, up to scaling, the equivalent denoising loss:

$$\mathbb{E}_{p(r)}\mathbb{E}_{p(\mathbf{y})}\mathbb{E}_{p_r(\mathbf{x}_r|\mathbf{y})}\left\|\hat{\mathbf{y}}_\phi(\mathbf{x}_r, r) - \mathbf{y}\right\|_2^2. \tag{17}$$

Diffusion models use an analogous reparameterization: the score estimator is $f_\phi(\mathbf{x}_\sigma, \sigma) = (\mathbf{x}_\sigma - \hat{\mathbf{y}}_\phi(\mathbf{x}_\sigma, \sigma))/\sigma$, leading to the analogous denoising objective

$$\mathbb{E}_{p(\sigma)}\mathbb{E}_{p(\mathbf{y})}\mathbb{E}_{p_\sigma(\mathbf{x}_\sigma|\mathbf{y})}\left\|\hat{\mathbf{y}}_\phi(\mathbf{x}_\sigma, \sigma) - \mathbf{y}\right\|_2^2, \tag{18}$$

with the corresponding distributions $p(\sigma)$ and $p_\sigma(\mathbf{x}_\sigma|\mathbf{y})$ replacing $p(r)$ and $p_r(\mathbf{x}_r|\mathbf{y})$.

As shown in §2.3, under the reparameterization $\sigma = r/\sqrt{D}$, the PFGM++ field estimator $f_\phi$ converges to its diffusion model counterpart as $D \to \infty$. Consequently, the PFGM++ denoising model $\hat{\mathbf{y}}_\phi(\mathbf{x}_r, r)$ also converges to the diffusion denoising model $\hat{\mathbf{y}}_\phi(\mathbf{x}_\sigma, \sigma)$.

## 3 Method

In this section, we present our **I**nverse **P**oisson **F**low **M**atching (IPFM), a distillation method for the recently proposed PFGM++ ODE-based generative framework. First, in §3.1, we formulate distillation as an Inverse Poisson Flow Matching Problem. Since this problem is not directly amenable to gradient-based optimization, we derive an equivalent, tractable objective that can be efficiently optimized with standard gradient-based methods. Second, in §3.2, we reveal a connection between our IPFM in the limit $D \to \infty$ and SiD (Zhou et al., 2024), a modern diffusion distillation method. Then, we show how to transfer SiD's practical improvement to our method in §3.3. Finally, in §3.4, we summarize our practical IPFM training algorithm for the single-step generator, and describe its extension to the multi-step setting. All proofs are provided in Appendix A.

## 3.1 The Inverse Poisson Flow Matching Problem

Let $f_\phi^*(\mathbf{x}_r, r)$ be the true renormalized Poisson flow obtained by applying the PFGM++ framework to a data distribution $p_{data}(\mathbf{y})$. Specifically, $f_\phi^*(\mathbf{x}_r, r)$ is a minimizer of the PFGM++ objective (12) and governs the electric dynamics (13), establishing a coupling between the prior $p_{r_{\max}}(\mathbf{x}_{r_{\max}})$ and the data distribution $p_{data}(\mathbf{y})$. We refer to this as the *teacher Poisson flow*.

Our goal is to distill the complex ODE-based sampling of the teacher Poisson flow into an efficient generator $G_\theta$ that requires only a few function evaluations. This generator defines a distribution $p_\theta(\mathbf{y})$. Applying the PFGM++ framework to $p_\theta(\mathbf{y})$ induces its own electrostatic field, which in turn yields a corresponding *student Poisson flow*, denoted $f_\psi(\mathbf{x}_r, r)$.

We formulate this distillation task as the *inverse Poisson flow matching problem*. The objective is to find a generator $G_\theta$ such that the electrostatic field induced by its output distribution $p_\theta(\mathbf{y})$ matches the field induced by the real data $p_{data}(\mathbf{y})$. Since the teacher flow $f_\phi^*$ and student flow $f_\psi$ are both derived from their respective fields, matching the fields ensures the flows are aligned. Figure 1 provides an overview of this problem. Mathematically, the problem is defined as follows:

**Definition 3.1** (Inverse Poisson Flow Matching Problem). *The inverse Poisson flow matching problem is defined as the following constrained optimization problem:*

$$\min_\theta \ \mathbb{E}_{r,\mathbf{x}_r} \left\| f_\psi(\mathbf{x}_r, r) - f_\phi^*(\mathbf{x}_r, r) \right\|_2^2, \quad s.t. \tag{19}$$

$$f_\psi = \arg\min_{f_\psi'} \mathbb{E}_{r,\mathbf{y},\mathbf{x}_r} \left\| f_\psi'(\mathbf{x}_r, r) - \frac{\mathbf{x}_r - \mathbf{y}}{r/\sqrt{D}} \right\|_2^2,$$

$$\mathbf{y} \sim p_\theta(\mathbf{y}), \quad \mathbf{x}_r \sim p_r(\mathbf{x}_r \mid \mathbf{y}), \quad r \sim \mathcal{U}[0, r_{\max}].$$

*Note that the minimization is over the weights of the few-step generator $G_\theta(\mathbf{y})$, which appears implicitly through its pushforward distribution $p_\theta(\mathbf{y})$.*

To justify this formulation, we now show that the inverse Poisson flow matching objective is well posed: its global optimum is achieved precisely when the generator distribution matches the real data distribution. This is established in the following theorem.

**Theorem 3.2** (Well-Posedness of the Inverse Poisson Flow Matching Problem). *Suppose the generator distribution $p_{\theta^*}$ is a global minimizer of the IPFM problem (3.1). Then, under mild regularity conditions on the densities, $p_{\theta^*}$ is equal to the data distribution $p_{data}$ (almost everywhere).*

However, directly optimizing objective (19) is intractable. The constraint requires solving an inner $\arg\min$ problem for $f_\psi$ that depends on the generator's distribution $p_\theta(\mathbf{y})$, making standard gradient-based optimization infeasible due to the need for backpropagation through this operation. To overcome this, we derive an equivalent but tractable objective:

**Theorem 3.3** (Tractable Reformulation of the Inverse Poisson Flow Matching Problem). *The constrained problem (19) is equivalent to the following unconstrained optimization problem for any positive weighting function $\lambda(r) > 0$:*

$$\min_\theta \max_\psi \mathbb{E}_{\substack{\mathbf{y} \sim p_\theta(\mathbf{y}), \\ r \sim \mathcal{U}[0, r_{\max}], \\ \mathbf{x}_r \sim p_r(\mathbf{x}_r | \mathbf{y})}} \lambda(r) \left[ \left\| f_\phi^*(\mathbf{x}_r, r) - \frac{\mathbf{x}_r - \mathbf{y}}{r/\sqrt{D}} \right\|_2^2 - \left\| f_\psi(\mathbf{x}_r, r) - \frac{\mathbf{x}_r - \mathbf{y}}{r/\sqrt{D}} \right\|_2^2 \right]. \tag{20}$$

Our tractable objective can be further reformulated using denoising models:

**Proposition 3.4** (Tractable Reformulation via Denoising Models). *The objective in Theorem 3.3 can be expressed in terms of denoising models as:*

$$\min_\theta \max_\psi \mathbb{E}_{\substack{\mathbf{y} \sim p_\theta(\mathbf{y}), \\ r \sim \mathcal{U}[0, r_{\max}], \\ \mathbf{x}_r \sim p_r(\mathbf{x}_r | \mathbf{y})}} \lambda(r) \left[ \left\| \hat{\mathbf{y}}_\phi^*(\mathbf{x}_r, r) - \mathbf{y} \right\|_2^2 - \left\| \hat{\mathbf{y}}_\psi(\mathbf{x}_r, r) - \mathbf{y} \right\|_2^2 \right]. \tag{21}$$

This minimax problem forms the foundation of our IPFM framework. Crucially, the inner maximization over $\psi$ is equivalent to training a student denoising model $\hat{\mathbf{y}}_\psi$ on the generator's distribution $p_\theta(\mathbf{y})$ using the standard PFGM++ objective (17), up to the weighting function $\lambda(r)$.

## 3.2 Connection to Score Identity Distillation

Several recent works (Zhou et al., 2024; Huang et al., 2024; Gushchin et al., 2025) have proposed related distillation approaches. Among them, Score Identity Distillation (SiD) focuses on distilling diffusion models by matching *score functions*, which stands in analogy to our objective of matching *electrostatic fields*. Given the established connection between PFGM++ and diffusion models in the $D \to \infty$ limit (see §2.3), we now investigate how our IPFM objective relates to SiD in this limit.

To formalize this connection, we begin by recalling our IPFM minimax objective from Eq. (21):

$$\mathcal{L}_{\text{IPFM}}^D = \mathbb{E}_{\substack{\mathbf{y} \sim p_\theta(\mathbf{y}) \\ r \sim \mathcal{U}[0, r_{\max}] \\ \mathbf{x}_r \sim p_r(\mathbf{x}_r | \mathbf{y})}} \lambda(r) \left[ \left\| \hat{\mathbf{y}}_\phi^*(\mathbf{x}_r, r) - \mathbf{y} \right\|_2^2 - \left\| \hat{\mathbf{y}}_\psi(\mathbf{x}_r, r) - \mathbf{y} \right\|_2^2 \right], \tag{22}$$

where the superscript $D$ emphasizes the auxiliary dimensionality parameter.

As established in §2.4, when $D \to \infty$ and $\sigma = r/\sqrt{D}$, the PFGM++ teacher model converges to a diffusion teacher model, and its perturbation kernel converges to the Gaussian kernel used in diffusion models. Consequently, it is natural to define the following asymptotic objective:

$$\mathcal{L}_{\text{IPFM}}^{D \to \infty} = \mathbb{E}_{\substack{\mathbf{y} \sim p_\theta(\mathbf{y}) \\ \sigma \sim \mathcal{U}[0, \sigma_{\max}] \\ \mathbf{x}_\sigma \sim p_\sigma(\mathbf{x}_\sigma | \mathbf{y})}} \lambda_\sigma(\sigma) \left[ \left\| \hat{\mathbf{y}}_\phi^*(\mathbf{x}_\sigma, \sigma) - \mathbf{y} \right\|_2^2 - \left\| \hat{\mathbf{y}}_\psi(\mathbf{x}_\sigma, \sigma) - \mathbf{y} \right\|_2^2 \right], \quad (23)$$

where $\mathbf{x}_\sigma \sim p_\sigma(\mathbf{x}_\sigma \mid \mathbf{y})$ is the diffusion perturbation kernel (15), and $\lambda_\sigma(\sigma)$ is a positive weighting function. Note that, up to the specific form of $\lambda_\sigma(\sigma)$, the inner maximization over $\psi$ corresponds to the standard diffusion training objective (18) applied to the generator's distribution.

We now recall the SiD *generator objective* (Zhou et al., 2024, Eq. 20):

$$\mathcal{L}_{\text{SiD}} = \mathbb{E}_{\substack{\mathbf{y} \sim p_\theta(\mathbf{y}) \\ \sigma \sim \mathcal{U}[0, \sigma_{\max}] \\ \mathbf{x}_\sigma \sim p_\sigma(\mathbf{x}_\sigma | \mathbf{y})}} \left[ \left\| \hat{\mathbf{y}}_\phi^*(\mathbf{x}_\sigma, \sigma) - \hat{\mathbf{y}}_\psi(\mathbf{x}_\sigma, \sigma) \right\|_2^2 \right.$$

$$\left. + \left\langle \hat{\mathbf{y}}_\phi^*(\mathbf{x}_\sigma, \sigma) - \hat{\mathbf{y}}_\psi(\mathbf{x}_\sigma, \sigma), \hat{\mathbf{y}}_\psi(\mathbf{x}_\sigma, \sigma) - \mathbf{y} \right\rangle \right], \quad (24)$$

where $\hat{\mathbf{y}}_\phi^*(\mathbf{x}_\sigma, \sigma)$ is the teacher diffusion model and $\hat{\mathbf{y}}_\psi(\mathbf{x}_\sigma, \sigma)$ is a student diffusion model trained on the generator's distribution $p_\theta(\mathbf{y})$.

The following proposition formalizes the relationship between these two objectives:

**Proposition 3.5** (Connection of our IPFM to SiD as $D \to \infty$). *The asymptotic IPFM generator objective (23) and the SiD objective (24) satisfy the following relation:*

$$\mathcal{L}_{IPFM}^{D \to \infty} = 2\mathcal{L}_{SiD} - \mathbb{E}_{\substack{\mathbf{y} \sim p_\theta(\mathbf{y}) \\ \sigma \sim \mathcal{U}[0, \sigma_{\max}] \\ \mathbf{x}_\sigma \sim p_\sigma(\mathbf{x}_\sigma | \mathbf{y})}} \lambda_\sigma(\sigma) \left\| \hat{\mathbf{y}}_\phi^*(\mathbf{x}_\sigma, \sigma) - \hat{\mathbf{y}}_\psi(\mathbf{x}_\sigma, \sigma) \right\|_2^2. \quad (25)$$

This close connection has a *significant practical implication*: it enables the direct transfer of hyper-parameters and techniques developed for SiD to our IPFM framework (see below).

### 3.3 SiD-Inspired Regularization

In practice SiD uses the following regularized objective (Zhou et al., 2024, Eq. 23):

$$\mathcal{L}_{\text{SiD}}^\alpha = \mathcal{L}_{\text{SiD}} - \alpha \mathbb{E}_{\substack{\mathbf{y} \sim p_\theta(\mathbf{y}) \\ \sigma \sim \mathcal{U}[0, \sigma_{\max}] \\ \mathbf{x}_\sigma \sim p_\sigma(\mathbf{x}_\sigma | \mathbf{y})}} \lambda_\sigma(\sigma) \left\| \hat{\mathbf{y}}_\phi^*(\mathbf{x}_\sigma, \sigma) - \hat{\mathbf{y}}_\psi(\mathbf{x}_\sigma, \sigma) \right\|_2^2. \quad (26)$$

From the relation (25), one may see that:

$$\mathcal{L}_{\text{SiD}}^{\alpha=\frac{1}{2}} = \mathcal{L}_{\text{SiD}} - \frac{1}{2} \mathbb{E}_{\sigma, \mathbf{y}, \mathbf{x}_\sigma} \lambda_\sigma(\sigma) \left\| \hat{\mathbf{y}}_\phi^*(\mathbf{x}_\sigma, \sigma) - \hat{\mathbf{y}}_\psi(\mathbf{x}_\sigma, \sigma) \right\|_2^2 = \frac{1}{2} \mathcal{L}_{\text{IPFM}}^{D \to \infty}. \quad (27)$$

This shows that $\mathcal{L}_{\text{SiD}}^{\alpha=\frac{1}{2}}$ is equivalent to $\mathcal{L}_{\text{IPFM}}^{D \to \infty}$. Given this, the regularization can be naturally transferred to our asymptotic objective (23):

$$\mathcal{L}_{\text{SiD}}^\alpha = \mathcal{L}_{\text{SiD}}^{\alpha=\frac{1}{2}} - (\alpha - 1/2)\mathbb{E}_{\sigma, \mathbf{y}, \mathbf{x}_\sigma} \lambda_\sigma(\sigma) \left\| \hat{\mathbf{y}}_\phi^*(\mathbf{x}_\sigma, \sigma) - \hat{\mathbf{y}}_\psi(\mathbf{x}_\sigma, \sigma) \right\|_2^2$$

$$= \frac{1}{2} \left[ \mathcal{L}_{\text{IPFM}}^{D \to \infty} - (2\alpha - 1)\mathbb{E}_{\sigma, \mathbf{y}, \mathbf{x}_\sigma} \lambda_\sigma(\sigma) \left\| \hat{\mathbf{y}}_\phi^*(\mathbf{x}_\sigma, \sigma) - \hat{\mathbf{y}}_\psi(\mathbf{x}_\sigma, \sigma) \right\|_2^2 \right] = \frac{1}{2} \mathcal{L}_{\text{IPFM}}^{\alpha, D \to \infty}.$$

This derivation yields two key insights. First, the SiD objective corresponds to our $\mathcal{L}_{\text{IPFM}}^{D \to \infty}$ framework with its regularization term scaled by $2\alpha - 1$ instead of $\alpha$. Second, this relationship provides a principled way to *extend the regularization to finite $D$*:

$$\mathcal{L}_{\text{IPFM}}^{\alpha, D} = \left[ \mathcal{L}_{\text{IPFM}}^D - (2\alpha - 1)\mathbb{E}_{\substack{\mathbf{y} \sim p_\theta(\mathbf{y}), \\ r \sim \mathcal{U}[0, r_{\max}], \\ \mathbf{x}_r \sim p_r(\mathbf{x}_r | \mathbf{y})}} \lambda(r) \left\| \hat{\mathbf{y}}_\phi^*(\mathbf{x}_r, r) - \hat{\mathbf{y}}_\psi(\mathbf{x}_r, r) \right\|_2^2 \right]. \quad (28)$$

Note that our original (unregularized) IPFM objective (21) is recovered when $\alpha = 0.5$.

Table 1: **Quantitative results of our IPFM distillation on CIFAR-10 and FFHQ 64x64.** We report FID scores for distilled generators across different auxiliary dimensions $D$, number of function evaluations (NFE), and regularization strengths $\alpha$. Note that (a) $\alpha = 0.5$ corresponds to our unregularized IPFM and (b) $D \to \infty$ with $\alpha = 1.0$ recovers SiD (the corresponding rows are highlighted). For comparison, we include the original PFGM++ teacher models evaluated in few-step regimes, demonstrating that distillation is essential for high-quality few-step generation. In each row (corresponding to a specific $D$), the best FID score is **bolded** and the second best is underlined.

| D | $\alpha$ | IPFM (ours) | | | Teacher (PFGM++, CIFAR-10) | | | | | |
|---|---|---|---|---|---|---|---|---|---|---|
| | | 1 | 2 | 4 | 1 | 5 | 9 | 17 | 25 | 35 |
| 128 | 0.5 | 5.38 | 2.68 | 2.08 | >100 | >100 | 37.79 | 3.32 | 2.07 | 1.92 |
| | 1.0 | 3.31 | 2.12 | **1.75** | | | | | | |
| 2048 | 0.5 | 5.47 | 3.05 | 2.02 | >100 | >100 | 37.14 | 3.37 | 2.03 | 1.91 |
| | 1.0 | 3.20 | 2.15 | **1.82** | | | | | | |
| $\infty$ (Diffusion) | 0.5 | 5.57 | 2.86 | 2.13 | >100 | >100 | 40.24 | 3.74 | 2.23 | 1.98 |
| | 1.0 | 3.47 | 2.15 | **1.86** | | | | | | |

| D | $\alpha$ | IPFM (ours) | | Teacher (PFGM++, FFHQ 64x64) | | | | | |
|---|---|---|---|---|---|---|---|---|---|
| | | 1 | 2 | 1 | 5 | 13 | 23 | 31 | 39 | 79 |
| 128 | 0.5 | 3.42 | 2.12 | >100 | >100 | 16.35 | 3.92 | 2.89 | 2.60 | 2.43 |
| | 1.0 | 2.40 | **1.72** | | | | | | | |
| $\infty$ (Diffusion) | 0.5 | 3.91 | 2.04 | >100 | >100 | 15.82 | 3.67 | 2.84 | 2.62 | 2.53 |
| | 1.0 | 2.60 | **1.70** | | | | | | | |

## 3.4 PRACTICAL IMPLEMENTATION

The practical implementation of the proposed IPFM method is summarized in Algorithm 1. The training procedure minimizes the objective (21) by iteratively performing two update steps: one for the student denoising model $\hat{\mathbf{y}}_\psi$ and one for the generator $G_\theta$. Our theoretical connection to SiD in §3.2 and §3.3 enables direct transfer of well-tuned hyperparameters from SiD to our method for the diffusion ($D \to \infty$) regime. Moreover, by following the PFGM++ (§2.3), we can extend these hyperparameters to finite values of $D$. Complete details are presented in Appendix B.

Although the distilled generator $G_\theta$ is capable of single-step sampling, its effective capacity can be enhanced by employing a multi-step sampling procedure. We construct this multi-step generator using the PFGM++ perturbation kernel (9), inspired by the approach of (Yin et al., 2024). The procedure employs a predetermined, monotonically decreasing noise-level schedule $\sigma_{\text{init}} = \sigma_0 > \sigma_1 > \ldots > \sigma_{N-1} > \sigma_N = \sigma_{\min}$, where $\sigma_{\text{init}}$ and $\sigma_{\min}$ are specified in Appendix B. This schedule is identical during both training and inference. The sampling process begins by sampling an initial point $\mathbf{x}_{r_0}$ from the prior distribution $p_{r_0}(\cdot \mid \mathbf{y} = \mathbf{0})$ at the highest noise level, $r_0 = \sigma_0 \sqrt{D}$. For each step $n = 0, 1, \ldots, N - 1$, the procedure consists of two operations: first, compute a denoised estimate $\hat{\mathbf{y}}_n = G_\theta(\mathbf{x}_{r_n}, \sigma_n)$; then, perturb this estimate with noise scaled to the next lower noise level, yielding $\mathbf{x}_{r_{n+1}} \sim p_{r_{n+1}}(\cdot \mid \hat{\mathbf{y}}_n)$, where $r_{n+1} = \sigma_{n+1} \sqrt{D}$. This cycle of denoising and perturbation repeats for $N$ steps. The final output is the denoised estimate from the last step, $\hat{\mathbf{y}}_N$. The complete procedure is detailed in Algorithm 2.

## 4 EXPERIMENTAL RESULTS

The goal of our experimental evaluation is to investigate two key aspects of the proposed method: (1) the sample quality achievable by models distilled with our IPFM, and (2) the impact of the auxiliary dimension $D$ on distillation performance. To this end, we benchmark our IPFM on all models provided by the PFGM++ authors, including both unconditional and class-conditional generation tasks. We perform a comprehensive analysis using one-, two-, and four-step generators, both with and without the SiD-inspired regularization introduced in §3.3. Each distillation run is performed with a fixed budget of 30k generator samples. This budget is selected due to computational con-

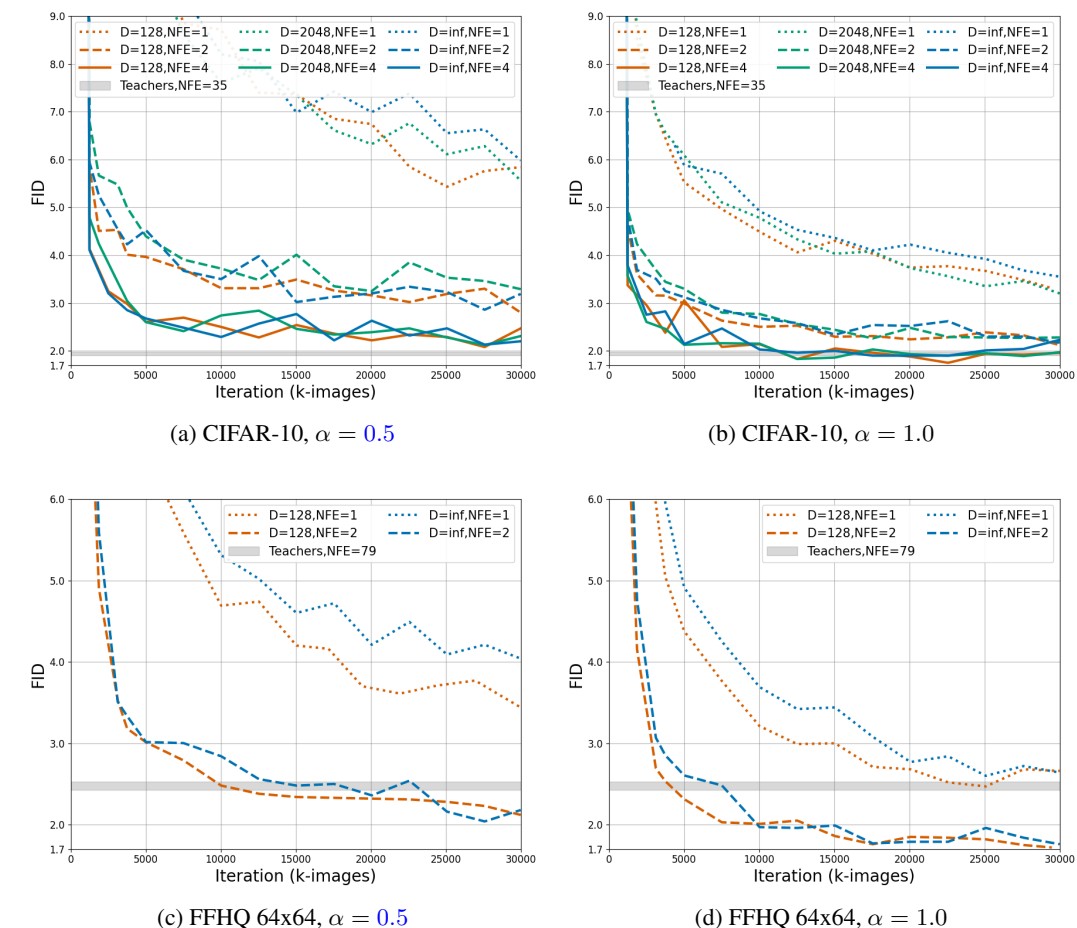

Figure 2: **Convergence of our IPFM.** Plots show FID evolution during training (measured in thousands of generator samples) for different IPFM settings. The gray horizontal band represents the range of final performance (minimum to maximum FID across different D) achieved by the original PFGM++ teacher models, providing a target quality threshold.

straints and proved sufficient for addressing our research questions. We use the Fréchet Inception Distance (FID) (Brock et al., 2018) to compare the model quality. More experimental details are presented in Appendix B, and additional results, including class-conditional generation, $\alpha$ ablation studies, and convergence analysis of teacher models, can be found in Appendix D. The code for experiments is based on PyTorch and will be made publicly available.

**Distillation Efficiency of our unregularized IPFM.** We first assess the core effectiveness of our IPFM framework without any additional regularization. The quantitative results, summarized in Table 1, demonstrate that our unregularized IPFM successfully distills the complex sampling process of PFGM++ into efficient few-step generators. Specifically, on CIFAR-10, a 4-step generator distilled with our IPFM matches the 35-NFE teacher's FID. On the more complex FFHQ 64x64 dataset, the improvement is even more pronounced: a 2-step generator is enough to surpass the sample quality of its 79-NFE teacher. This demonstrates that our method effectively solves the distillation problem, validating our core theoretical formulation. The qualitative results are presented in Appendix E.

**Enhancing our IPFM with SiD-Inspired Regularization.** Motivated by the connection revealed in §3.2 and §3.3, we incorporated SiD-inspired regularization into our framework. Based on an ablation study (see Appendix B), we selected $\alpha = 1.0$ as it provides a stable and effective balance across all teacher architectures and values of $D$. Table 1 confirms the high effectiveness of this regularization. Our IPFM with SiD-inspired regularization not only accelerates convergence but also

elevates the final performance of the distilled models. With this enhancement, our 4-step generator on CIFAR-10 and 2-step generator on FFHQ 64x64 now clearly outperform their respective teacher models. This confirms that the practical benefits of SiD transfer effectively to the broader electrostatic setting through our unified framework. The qualitative results are presented in Appendix E.

**The Impact of Auxiliary Dimension $D$.** As shown in Figure 2, our IPFM converges faster for smaller values of $D$ compared to the diffusion setting ($D \to \infty$), which, when equipped with the regularization introduced in §3.3, corresponds exactly to SiD. We attribute this to the heavier-tailed perturbation kernels and enhanced robustness of teacher models at lower $D$. During distillation, the teacher model is evaluated on the perturbed generator outputs. Early in training, the generator produces samples far from the real data distribution; even after perturbation, these remain out-of-distribution for the teacher, leading to a noisy training signal. This issue is mitigated at lower $D$ for two reasons: (1) Heavier-tailed kernels better conceal the discrepancy between the generator's distribution and the true data distribution, and (2) teacher models with smaller $D$ exhibit greater robustness to distribution shift, as demonstrated in the original PFGM++ study. The effect is most noticeable in the single-step generation setting. With more sampling steps, the generator rapidly produces high-quality samples, reducing the impact of this phenomenon on the considered datasets.

This finding has an **important implication**. Since our method with $D \to \infty$ and $\alpha = 1.0$ is equivalent to SiD (§3.3), which produces one-step generators that outperform their teachers, we conclude that our framework is at least as effective. The superior convergence of finite $D$ suggests that our IPFM can achieve similar or better performance more efficiently.

## 5 DISCUSSION

**Baselines.** To the best of our knowledge, SiD is the only GAN-free distillation method that can match (or even surpass) the performance of a diffusion (the $D \to \infty$ regime of PFGM++) teacher. Our results show that finite-$D$ variants of our IPFM converge at least as well as, and in some cases better than, this diffusion limit. This indicates that our IPFM can attain teacher-level performance in a single step given sufficient compute. By contrast, other GAN-free distillation approaches (e.g., consistency-based models) have not been demonstrated to reach teacher performance even for diffusion teachers. Moreover, there are no specific adaptations of these approaches for PFGM++ teachers on standard generative benchmarks (the only adaptation is (Hein et al., 2025) , which focuses on solving a low-dose CT image denoising task). For this reason, we do not include other baselines for finite-$D$ Poisson-flow distillation and focus our evaluation on the theoretically grounded connection between our IPFM and SiD across different values of $D$. That said, in Appendix D.1 we provide a sanity-check experiment with a finite-$D$ PFGM++ showing that our IPFM significantly surpasses Consistency Trajectory Models (Kim et al., 2023), an advanced consistency-based distillation.

**Potential Impact.** Our distillation method applies to all auxiliary dimensions $D$ of the electrostatic generative framework PFGM++ and achieves teacher-level quality with just one or a few function evaluations. Moreover, we find that PFGM++ models with finite $D$ are more amenable to our distillation, highlighting their superior generative properties compared to their diffusion counterparts ($D \to \infty$). We believe our findings not only enable efficient training of high-performance distilled models but also underscore the broader potential of electrostatic generative models, revealing unique optimization advantages beyond the diffusion paradigm.

**Limitations.** Our method requires solving a minimax optimization problem that alternates between updating an auxiliary student model and the generator, which is computationally expensive. Each generator update also requires backpropagation through both the teacher and student models, resulting in memory requirements approximately three times higher than training the teacher alone. However, this limitation is shared among other modern distillation techniques (Zhou et al., 2024; Huang et al., 2024; Gushchin et al., 2025).

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

# A PROOFS

**Theorem 3.2** (Well-Posedness of the Inverse Poisson Flow Matching Problem). *Suppose the generator distribution $p_{\theta^*}$ is a global minimizer of the IPFM problem (3.1). Then, under mild regularity conditions on the densities, $p_{\theta^*}$ is equal to the data distribution $p_{data}$ (almost everywhere).*

*Proof.* For convenience, we view the IPFM objective as a functional of the generator density $p$ (for a parameter $\theta$, we write $p = p_\theta$) and define

$$L(p) = \mathbb{E}_{r,\mathbf{x}_r} \big\| f_p(\mathbf{x}_r, r) - f_{p_{data}}(\mathbf{x}_r, r) \big\|^2, \tag{29}$$

where

- $f_p(\mathbf{x}_r, r)$ is the (student) Poisson flow induced by the generator density $p$,
- $f_{p_{data}}(\mathbf{x}_r, r)$ is the (teacher) Poisson flow induced by the data density $p_{data}$,
- $r \sim \mathcal{U}[0, r_{max}]$ and $\mathbf{x}_r \sim p_r(\mathbf{x}_r)$, where $p_r(\mathbf{x}_r) = \int p_r(\mathbf{x}_r \mid \mathbf{y}) \, p(\mathbf{y}) \, d\mathbf{y}$.

Recall that for any density $p$, the normalized Poisson flow is defined as the (unique) minimizer of the PFGM++ denoising objective (12):

$$f_p(\mathbf{x}_r, r) = \arg\min_{f'} \mathbb{E}\left[ \left\| f'(\mathbf{x}_r, r) - \frac{\mathbf{x}_r - \mathbf{y}}{r/\sqrt{D}} \right\|^2 \right] = \mathbb{E}\left[ \frac{\mathbf{x}_r - \mathbf{y}}{r/\sqrt{D}} \,\bigg|\, \mathbf{x}_r, r \right]_p, \tag{30}$$

i.e., the conditional expectation under the joint law over $(r, \mathbf{y}, \mathbf{x}_r)$ induced by $p$.

Moreover, PFGM++ (Xu et al., 2023, Proposition 3.2) relates this normalized flow to the augmented electrostatic field $\mathbf{E}_p$ and the marginal density $p_r$:

$$(f_p(\mathbf{x}_r, r), r) \propto \frac{S_{N+D-1}}{p_r(\mathbf{x}_r)} \mathbf{E}_p(\mathbf{x}_r, r). \tag{31}$$

Given this, our proof consists of the following four steps:

1. Any global minimizer $p^*$ satisfies $L(p^*) = 0$;
2. $L(p^*) = 0$ implies $f_{p^*} = f_{p_{data}}$ almost everywhere;
3. $f_{p_1} = f_{p_2}$ implies $p_{1,r} = p_{2,r} \ \forall r > 0$;
4. If $p_{1,r_0} = p_{2,r_0}$ for some fixed $r_0 > 0$, then $p_1 = p_2$.

Combining these four steps, we obtain the following chain of implications:

1. If $p^* = p_{\theta^*}$ is a global minimizer, then $L(p_{\theta^*}) = 0$ (Step 1);
2. $L(p_{\theta^*}) = 0$ implies $f_{p_{\theta^*}} = f_{p_{data}}$ almost everywhere (Step 2);
3. $f_{p_{\theta^*}} = f_{p_{data}}$ implies $p_{\theta^*, r} = p_{data, r}$ for all $r > 0$ (Step 3);
4. In particular, $p_{\theta^*, r_0} = p_{data, r_0}$ for some fixed $r_0 > 0$ implies $p_{\theta^*} = p_{data}$ almost everywhere (Step 4).

Thus any global minimizer $p_{\theta^*}$ of the IPFM objective must coincide with the data distribution $p_{data}$ (almost everywhere), which proves the theorem. It remains to derive the basic steps one by one.

**Step 1: Any global minimizer $p^*$ satisfies $L(p^*) = 0$.**

*Proof.* By definition, $L(p) \geq 0$ for all $p$ since the integrand in (29) is a squared norm.

If $p = p_{data}$, then

$$f_p(\mathbf{x}_r, r) = \mathbb{E}\left[ \frac{\mathbf{x}_r - \mathbf{y}}{r/\sqrt{D}} \,\bigg|\, \mathbf{x}_r, r \right]_p = \mathbb{E}\left[ \frac{\mathbf{x}_r - \mathbf{y}}{r/\sqrt{D}} \,\bigg|\, \mathbf{x}_r, r \right]_{p_{data}} = f_{p_{data}}(\mathbf{x}_r, r),$$

for all $(\mathbf{x}_r, r)$, by uniqueness of the PFGM++ minimizer (30). Plugging this into (29) yields $L(p_{data}) = 0$. Since $L(p) \geq 0$ for all $p$, it follows that $\inf_p L(p) = 0$. If $p^*$ is a global minimizer of $L$, then necessarily $L(p^*) = \inf_p L(p) = 0$, which completes Step 1.

**Step 2:** $L(p^*) = 0$ **implies** $f_{p^*} = f_{p_{data}}$ **almost everywhere.**

*Proof.* For any $r > 0$, the marginal density $p_r^*$ is positive everywhere on $\mathbb{R}^N$ due to underlying PFGM++ perturbation kernel is positive everywhere on $\mathbb{R}^N$. Together with the nonnegativity of the objective $L(p) \geq 0$, the condition $L(p^*) = 0$ implies that the integrand in (29) is zero almost everywhere. Hence, $f_{p^*} = f_{p_{data}}$ almost everywhere, which completes Step 2.

**Step 3: Equality of normalized flows implies equality of marginals.**

*Proof.* By (31), we have

$$(f_{p_i}(\mathbf{x}_r, r), r) = \frac{S_{N+D-1}}{p_{i,r}(\mathbf{x}_r)} \mathbf{E}^{p_i}(\mathbf{x}_r, r), \qquad i \in \{1, 2\},$$

where $p_{i,r}$ are the corresponding marginals and $\mathbf{E}^{p_i}$ the augmented fields.

Using $f_{p_1} = f_{p_2}$, we obtain

$$\frac{S_{N+D-1}}{p_{1,r}(\mathbf{x}_r)} \mathbf{E}^{p_1}(\mathbf{x}_r, r) = \frac{S_{N+D-1}}{p_{2,r}(\mathbf{x}_r)} \mathbf{E}^{p_2}(\mathbf{x}_r, r),$$

hence

$$\mathbf{E}^{p_1}(\mathbf{x}_r, r) = a(\mathbf{x}_r, r) \mathbf{E}^{p_2}(\mathbf{x}_r, r), \qquad a(\mathbf{x}_r, r) := \frac{p_{1,r}(\mathbf{x}_r)}{p_{2,r}(\mathbf{x}_r)}. \tag{32}$$

Further recall from physics that the electric field is a gradient field:

$$\mathbf{E}^p(\mathbf{x}_r, r) = -\nabla \Phi^p(\mathbf{x}_r, r),$$

where

$$\Phi^p(\mathbf{x}_r, r) \propto \int \frac{1}{\left(\|\mathbf{x}_r - \mathbf{y}\|_2^2 + r^2\right)^{\frac{N+D-2}{2}}} p(\mathbf{y}) \, d\mathbf{y}$$

is a potential function and the gradient as taken over $(\mathbf{x}_r, r)$ and the operator $\nabla$ represents the gradient taken with respect to the variables $(\mathbf{x}_r, r)$. If $\Phi^p$ is twice continuously differentiable, it's mixed partial derivatives commute:

$$\partial_{x_k} \mathbf{E}_l^p = -\partial_{x_k, x_l}^2 \Phi^p = -\partial_{x_l, x_k}^2 \Phi^p = \partial_{x_l} \mathbf{E}_k^p, \tag{33}$$

where $\partial_{x_k}$ denotes differentiation with respect to the $k$-th spatial (or radial) coordinate and $\mathbf{E}_k^p$ denotes the $k$-th component of the vector $\mathbf{E}^p$.

Apply (33) to $p = p_1$ and use the relation (32). For any indices $k, l$, we have

$$\partial_{x_k} \mathbf{E}_l^{p_1} = \partial_{x_k} \left(a \mathbf{E}_l^{p_2}\right) = (\partial_{x_k} a) \mathbf{E}_l^{p_2} + a \, \partial_{x_k} \mathbf{E}_l^{p_2},$$
$$\partial_{x_l} \mathbf{E}_k^{p_1} = \partial_{x_l} \left(a \mathbf{E}_k^{p_2}\right) = (\partial_{x_l} a) \mathbf{E}_k^{p_2} + a \, \partial_{x_l} \mathbf{E}_k^{p_2}.$$

By (33), the left-hand sides are equal, and since $\mathbf{E}^{p_2}$ is also a gradient field its mixed partials commute, so the the right terms cancel. We obtain

$$(\partial_{x_k} a) \mathbf{E}_l^{p_2} = (\partial_{x_l} a) \mathbf{E}_k^{p_2} \qquad \text{for all } k, l.$$

For $r > 0$, the last component (with index $N+1$ and corresponding $r$) of the field is always positive. Then, for all $k$,

$$(\partial_{x_k} a) = \frac{\partial_{x_{N+1}} a}{\mathbf{E}_{N+1}^{p_2}} \mathbf{E}_k^{p_2}.$$

That is, there exists a scalar function $s := (\partial_{x_{N+1}} a) / \mathbf{E}_{N+1}^{p_2}$ such that

$$\nabla a(\mathbf{x}_r, r) = s(\mathbf{x}_r, r) \mathbf{E}^{p_2}(\mathbf{x}_r, r). \tag{34}$$

Next, recall that in the charge-free region $r > 0$ the electric field is divergence-free:

$$\nabla \cdot \mathbf{E}^p = 0.$$

Using $\mathbf{E}^{p_1} = a\mathbf{E}^{p_2}$ from (32) we have

$$0 = \nabla \cdot \mathbf{E}^{p_1} = \nabla \cdot (a\mathbf{E}^{p_2}) = \nabla a \cdot \mathbf{E}^{p_2} + a\nabla \cdot \mathbf{E}^{p_2} = \nabla a \cdot \mathbf{E}^{p_2}, \tag{35}$$

since $\nabla \cdot \mathbf{E}^{p_2} = 0$ as well. Combining (34) and (35) gives

$$\nabla a(\mathbf{x}_r, r) \cdot \mathbf{E}^{p_2}(\mathbf{x}_r, r) = s(\mathbf{x}_r, r) \left\|\mathbf{E}^{p_2}(\mathbf{x}_r, r)\right\|_2^2 = 0.$$

This implies $s(\mathbf{x}_r, r) = 0$ and hence

$$\nabla a(\mathbf{x}_r, r) = 0 \quad \Rightarrow \quad a(\mathbf{x}_r, r) = \text{const.}$$

Thus, from (32),

$$\frac{p_{1,r}(\mathbf{x}_r)}{p_{2,r}(\mathbf{x}_r)} = a(\mathbf{x}_r, r) = c,$$

where $c \in \mathbb{R}$ is some constant which does not depend on $\mathbf{x}_r$ and $r$.

Finally, both $p_{1,r}$ and $p_{2,r}$ are normalized densities:

$$1 = \int p_{1,r}(\mathbf{x}_r) \, d\mathbf{x}_r = c \cdot \int p_{2,r}(\mathbf{x}_r) \, d\mathbf{x}_r = c,$$

so we conclude that $c \equiv 1$. Therefore,

$$p_{1,r}(\mathbf{x}_r) = p_{2,r}(\mathbf{x}_r) \quad \text{for all } r > 0, \text{ almost every } \mathbf{x}_r.$$

This completes Step 3.

**Step 4: Equality of all marginals implies equality of base distributions.**

*Proof.* By the PFGM++ construction, the marginal $p_r$ can be written as a convolution of the base density $p$ with the PFGM++ kernel $K_r$ (see (9)):

$$p_r(\mathbf{x}_r) = (K_r * p)(\mathbf{x}_r) := \int_{\mathbb{R}^N} K_r(\mathbf{x}_r - \mathbf{y}) \, p(\mathbf{y}) \, d\mathbf{y}.$$

Let $p_1, p_2$ be two densities such that $p_{1,r} = p_{2,r}$ for all $r > 0$. Fix any $r_0 > 0$. Then

$$K_{r_0} * p_1 = K_{r_0} * p_2.$$

Taking Fourier transforms (characteristic functions) yields

$$\widehat{p_{1,r_0}}(\boldsymbol{\xi}) = \widehat{K_{r_0}}(\boldsymbol{\xi})\,\widehat{p_1}(\boldsymbol{\xi}), \qquad \widehat{p_{2,r_0}}(\boldsymbol{\xi}) = \widehat{K_{r_0}}(\boldsymbol{\xi})\,\widehat{p_2}(\boldsymbol{\xi}),$$

for all $\boldsymbol{\xi} \in \mathbb{R}^N$, where $\widehat{\cdot}$ denotes the Fourier transform. Since $p_{1,r_0} = p_{2,r_0}$, we have

$$\widehat{K_{r_0}}(\boldsymbol{\xi})\,\widehat{p_1}(\boldsymbol{\xi}) = \widehat{K_{r_0}}(\boldsymbol{\xi})\,\widehat{p_2}(\boldsymbol{\xi}) \quad \text{for all } \boldsymbol{\xi}.$$

The Fourier transform $\widehat{K_{r_0}}(\boldsymbol{\xi})$ of the PFGM++ kernel is strictly positive (and hence non-zero) for all $\boldsymbol{\xi}$. Therefore,

$$\widehat{p_1}(\boldsymbol{\xi}) = \widehat{p_2}(\boldsymbol{\xi}) \quad \text{for all } \boldsymbol{\xi}.$$

By uniqueness of characteristic functions, this implies $p_1 = p_2$ almost everywhere.

This completes Step 4.

**Theorem 3.3** (Tractable Reformulation of the Inverse Poisson Flow Matching Problem). *The constrained problem (19) is equivalent to the following unconstrained optimization problem for any positive weighting function $\lambda(r) > 0$:*

$$\min_{\theta} \max_{\psi} \mathbb{E}_{\substack{\mathbf{y} \sim p_\theta(\mathbf{y}), \\ r \sim \mathcal{U}[0, r_{\max}], \\ \mathbf{x}_r \sim p_r(\mathbf{x}_r|\mathbf{y})}} \lambda(r) \left[ \left\| f_\phi^*(\mathbf{x}_r, r) - \frac{\mathbf{x}_r - \mathbf{y}}{r/\sqrt{D}} \right\|_2^2 - \left\| f_\psi(\mathbf{x}_r, r) - \frac{\mathbf{x}_r - \mathbf{y}}{r/\sqrt{D}} \right\|_2^2 \right].$$

*Proof.* Consider the inverse Poisson flow matching problem:

$$\min_{\theta} \ \mathbb{E}_{r,\mathbf{x}_r} \left\| f_\psi(\mathbf{x}_r, r) - f_\phi^*(\mathbf{x}_r, r) \right\|_2^2, \quad \text{s.t.} \tag{36}$$

$$f_\psi = \arg\min_{f'_\psi} \mathbb{E}_{r,\mathbf{y},\mathbf{x}_r} \left\| f'_\psi(\mathbf{x}_r, r) - \frac{\mathbf{x}_r - \mathbf{y}}{r/\sqrt{D}} \right\|_2^2, \tag{37}$$

$$\mathbf{y} \sim p_\theta(\mathbf{y}), \quad \mathbf{x}_r \sim p_r(\mathbf{x}_r \mid \mathbf{y}), \quad r \sim \mathcal{U}[0, r_{\max}].$$

**Equivalent weighted problem.** To begin with, note that for any positive weighting function $\lambda(r) > 0$, the solution for the weighted constraint (37) remains the same and equals

$$f_\psi(\mathbf{x}_r, r) = \arg\min_{f'_\psi} \mathbb{E}_{r,\mathbf{y},\mathbf{x}_r} \lambda(r) \left\| f'_\psi(\mathbf{x}_r, r) - \frac{\mathbf{x}_r - \mathbf{y}}{r/\sqrt{D}} \right\|_2^2 = \mathbb{E}_{\mathbf{y}|\mathbf{x}_r, r} \left[ \frac{\mathbf{x}_r - \mathbf{y}}{r/\sqrt{D}} \right]. \tag{38}$$

Additionally, we can introduce a positive weighting function $\lambda(r) > 0$ to the main functional (36):

$$\mathbb{E}_{r,\mathbf{x}_r} \lambda(r) \left\| f_\psi(\mathbf{x}_r, r) - f_\phi^*(\mathbf{x}_r, r) \right\|_2^2,$$

since it does not change the optimum value (which equals 0) and this optimum is reached only when $f_\psi(\mathbf{x}_r, r) \equiv f_\phi^*(\mathbf{x}_r, r)$.

Therefore, we can equivalently consider the following problem:

$$\min_{\theta} \ \mathbb{E}_{r,\mathbf{x}_r} \lambda(r) \left\| f_\psi(\mathbf{x}_r, r) - f_\phi^*(\mathbf{x}_r, r) \right\|_2^2, \quad \text{s.t.} \tag{39}$$

$$f_\psi = \arg\min_{f'_\psi} \mathbb{E}_{r,\mathbf{y},\mathbf{x}_r} \lambda(r) \left\| f'_\psi(\mathbf{x}_r, r) - \frac{\mathbf{x}_r - \mathbf{y}}{r/\sqrt{D}} \right\|_2^2, \tag{40}$$

$$\mathbf{y} \sim p_\theta(\mathbf{y}), \quad \mathbf{x}_r \sim p_r(\mathbf{x}_r \mid \mathbf{y}), \quad r \sim \mathcal{U}[0, r_{\max}].$$

**Tractable reformulation.** Now let's reformulate the obtained weighted inverse Poisson flow problem in a tractable way.

To begin with, let's rewrite the main objective (39):

$$\mathbb{E}_{r,\mathbf{x}_r} \lambda(r) \left\| f_\psi(\mathbf{x}_r, r) - f_\phi^*(\mathbf{x}_r, r) \right\|_2^2 =$$

$$\mathbb{E}_{r,\mathbf{x}_r} \lambda(r) \| f_\psi(\mathbf{x}_r, r) \|_2^2 - 2\mathbb{E}_{r,\mathbf{x}_r} \lambda(r) \langle f_\psi(\mathbf{x}_r, r), f_\phi^*(\mathbf{x}_r, r) \rangle + \mathbb{E}_{r,\mathbf{x}_r} \lambda(r) \| f_\phi^*(\mathbf{x}_r, r) \|_2^2 \stackrel{(38)}{=}$$

$$\mathbb{E}_{r,\mathbf{x}_r} \lambda(r) \| f_\psi(\mathbf{x}_r, r) \|_2^2 - 2\mathbb{E}_{r,\mathbf{x}_r} \lambda(r) \left\langle \mathbb{E}_{\mathbf{y}|\mathbf{x}_r,r} \left[ \frac{\mathbf{x}_r - \mathbf{y}}{r/\sqrt{D}} \right], f_\phi^*(\mathbf{x}_r, r) \right\rangle + \mathbb{E}_{r,\mathbf{x}_r} \lambda(r) \| f_\phi^*(\mathbf{x}_r, r) \|_2^2 =$$

$$\mathbb{E}_{r,\mathbf{x}_r} \lambda(r) \| f_\psi(\mathbf{x}_r, r) \|_2^2 + \mathbb{E}_{r,\mathbf{y},\mathbf{x}_r} \lambda(r) \left[ -2 \left\langle \left[ \frac{\mathbf{x}_r - \mathbf{y}}{r/\sqrt{D}} \right], f_\phi^*(\mathbf{x}_r, r) \right\rangle + \| f_\phi^*(\mathbf{x}_r, r) \|_2^2 \right] \pm \mathbb{E}_{r,\mathbf{y},\mathbf{x}_r} \lambda(r) \left\| \frac{\mathbf{x}_r - \mathbf{y}}{r/\sqrt{D}} \right\|_2^2 =$$

$$\mathbb{E}_{r,\mathbf{x}_r} \lambda(r) \| f_\psi(\mathbf{x}_r, r) \|_2^2 - \mathbb{E}_{r,\mathbf{y},\mathbf{x}_r} \lambda(r) \left\| \frac{\mathbf{x}_r - \mathbf{y}}{r/\sqrt{D}} \right\|_2^2 + \mathbb{E}_{r,\mathbf{y},\mathbf{x}_r} \lambda(r) \left\| f_\phi^*(\mathbf{x}_r, r) - \frac{\mathbf{x}_r - \mathbf{y}}{r/\sqrt{D}} \right\|_2^2 \tag{41}$$

One may note that the only intractable term here is $\mathbb{E}_{r,\mathbf{x}_r}\lambda(r)\|f_\psi(\mathbf{x}_r,r)\|_2^2$. To circumvent this challenge we rewrite the constraint (40):

$$\min_{f'_\psi}\mathbb{E}_{r,\mathbf{y},\mathbf{x}_r}\lambda(r)\left\|f'_\psi(\mathbf{x}_r,r)-\frac{\mathbf{x}_r-\mathbf{y}}{r/\sqrt{D}}\right\|_2^2=$$

$$\mathbb{E}_{r,\mathbf{y},\mathbf{x}_r}\lambda(r)\left\|f_\psi(\mathbf{x}_r,r)-\frac{\mathbf{x}_r-\mathbf{y}}{r/\sqrt{D}}\right\|_2^2=$$

$$\mathbb{E}_{r,\mathbf{x}_r}\lambda(r)\|f_\psi(\mathbf{x}_r,r)\|_2^2-2\mathbb{E}_{r,\mathbf{y},\mathbf{x}_r}\lambda(r)\left\langle f_\psi(\mathbf{x}_r,r),\frac{\mathbf{x}_r-\mathbf{y}}{r/\sqrt{D}}\right\rangle+\mathbb{E}_{r,\mathbf{y},\mathbf{x}_r}\lambda(r)\left\|\frac{\mathbf{x}_r-\mathbf{y}}{r/\sqrt{D}}\right\|_2^2=$$

$$\mathbb{E}_{r,\mathbf{x}_r}\|f_\psi(\mathbf{x}_r,r)\|_2^2-2\mathbb{E}_{r,\mathbf{x}_r}\lambda(r)\left\langle f_\psi(\mathbf{x}_r,r),\mathbb{E}_{\mathbf{y}|\mathbf{x}_r,r}\frac{\mathbf{x}_r-\mathbf{y}}{r/\sqrt{D}}\right\rangle+\mathbb{E}_{r,\mathbf{y},\mathbf{x}_r}\lambda(r)\left\|\frac{\mathbf{x}_r-\mathbf{y}}{r/\sqrt{D}}\right\|_2^2\overset{(38)}{=}$$

$$\mathbb{E}_{r,\mathbf{x}_r}\|f_\psi(\mathbf{x}_r,r)\|_2^2-2\mathbb{E}_{r,\mathbf{x}_r}\lambda(r)\langle f_\psi(\mathbf{x}_r,r),f_\psi(\mathbf{x}_r,r)\rangle+\mathbb{E}_{r,\mathbf{y},\mathbf{x}_r}\lambda(r)\left\|\frac{\mathbf{x}_r-\mathbf{y}}{r/\sqrt{D}}\right\|_2^2=$$

$$\mathbb{E}_{r,\mathbf{x}_r}\|f_\psi(\mathbf{x}_r,r)\|_2^2-2\mathbb{E}_{r,\mathbf{x}_r}\lambda(r)\|f_\psi(\mathbf{x}_r,r)\|_2^2+\mathbb{E}_{r,\mathbf{y},\mathbf{x}_r}\lambda(r)\left\|\frac{\mathbf{x}_r-\mathbf{y}}{r/\sqrt{D}}\right\|_2^2=$$

$$-\mathbb{E}_{r,\mathbf{x}_r}\lambda(r)\|f_\psi(\mathbf{x}_r,r)\|_2^2+\mathbb{E}_{r,\mathbf{y},\mathbf{x}_r}\lambda(r)\left\|\frac{\mathbf{x}_r-\mathbf{y}}{r/\sqrt{D}}\right\|_2^2.$$

Therefore, we can express the intractable term in the following way:

$$\mathbb{E}_{r,\mathbf{x}_r}\lambda(r)\|f_\psi(\mathbf{x}_r,r)\|_2^2=\mathbb{E}_{r,\mathbf{y},\mathbf{x}_r}\lambda(r)\left\|\frac{\mathbf{x}_r-\mathbf{y}}{r/\sqrt{D}}\right\|_2^2-\min_{f'_\psi}\mathbb{E}_{r,\mathbf{y},\mathbf{x}_r}\lambda(r)\left\|f'_\psi(\mathbf{x}_r,r)-\frac{\mathbf{x}_r-\mathbf{y}}{r/\sqrt{D}}\right\|_2^2$$

$$(42)$$

Substituting (42) into the reformulated main objective (41) provides us with desired result:

$$\mathbb{E}_{r,\mathbf{x}_r}\lambda(r)\|f_\psi(\mathbf{x}_r,r)-f_\phi(\mathbf{x}_r,r)\|_2^2=$$

$$\mathbb{E}_{r,\mathbf{y},\mathbf{x}_r}\lambda(r)\left\|\frac{\mathbf{x}_r-\mathbf{y}}{r/\sqrt{D}}\right\|_2^2-\mathbb{E}_{r,\mathbf{y},\mathbf{x}_r}\lambda(r)\left\|\frac{\mathbf{x}_r-\mathbf{y}}{r/\sqrt{D}}\right\|_2^2+\mathbb{E}_{r,\mathbf{y},\mathbf{x}_r}\lambda(r)\left\|f_\phi^*(\mathbf{x}_r,r)-\frac{\mathbf{x}_r-\mathbf{y}}{r/\sqrt{D}}\right\|_2^2$$

$$-\min_{f'_\psi}\mathbb{E}_{r,\mathbf{y},\mathbf{x}_r}\lambda(r)\left\|f'_\psi(\mathbf{x}_r,r)-\frac{\mathbf{x}_r-\mathbf{y}}{r/\sqrt{D}}\right\|_2^2=$$

$$\mathbb{E}_{r,\mathbf{y},\mathbf{x}_r}\lambda(r)\left\|f_\phi^*(\mathbf{x}_r,r)-\frac{\mathbf{x}_r-\mathbf{y}}{r/\sqrt{D}}\right\|_2^2-\min_{f'_\psi}\mathbb{E}_{r,\mathbf{y},\mathbf{x}_r}\lambda(r)\left\|f'_\psi(\mathbf{x}_r,r)-\frac{\mathbf{x}_r-\mathbf{y}}{r/\sqrt{D}}\right\|_2^2.$$

**Proposition 3.4** (Tractable Reformulation via Denoising Models). *The objective in Theorem 3.3 can be expressed in terms of denoising models as:*

$$\min_\theta\max_\psi\mathop{\mathbb{E}}_{\substack{\mathbf{y}\sim p_\theta(\mathbf{y}),\\r\sim\mathcal{U}[0,r_{\max}],\\\mathbf{x}_r\sim p_r(\mathbf{x}_r|\mathbf{y})}}\lambda(r)\left[\|\hat{\mathbf{y}}_\phi^*(\mathbf{x}_r,r)-\mathbf{y}\|_2^2-\|\hat{\mathbf{y}}_\psi(\mathbf{x}_r,r)-\mathbf{y}\|_2^2\right].$$

*Proof.* Consider the reformulated tractable objective from Theorem 3.3:

$$\min_\theta\max_\psi\mathop{\mathbb{E}}_{\substack{\mathbf{y}\sim p_\theta(\mathbf{y}),\\r\sim\mathcal{U}[0,r_{\max}],\\\mathbf{x}_r\sim p_r(\mathbf{x}_r|\mathbf{y})}}\lambda(r)\left[\left\|f_\phi^*(\mathbf{x}_r,r)-\frac{\mathbf{x}_r-\mathbf{y}}{r/\sqrt{D}}\right\|_2^2-\left\|f_\psi(\mathbf{x}_r,r)-\frac{\mathbf{x}_r-\mathbf{y}}{r/\sqrt{D}}\right\|_2^2\right].$$

Recall that the Poisson flow $f(\mathbf{x}_r,r)$ can be recovered from the denoising model $\hat{\mathbf{y}}(\mathbf{x}_r,r)$ via:

$$f(\mathbf{x}_r,r)=\frac{\mathbf{x}_r-\hat{\mathbf{y}}(\mathbf{x}_r,r)}{r/\sqrt{D}}.$$

Substituting this relationship into the objective yields:

$$\min_\theta \max_\psi \mathbb{E}_{\substack{\mathbf{y}\sim p_\theta(\mathbf{y}),\\ r\sim\mathcal{U}[0,r_{\max}],\\ \mathbf{x}_r\sim p_r(\mathbf{x}_r|\mathbf{y})}} \frac{\lambda(r)}{r^2/D}\left[\left\|\hat{\mathbf{y}}_\phi^*(\mathbf{x}_r,r)-\mathbf{y}\right\|_2^2-\left\|\hat{\mathbf{y}}_\psi(\mathbf{x}_r,r)-\mathbf{y}\right\|_2^2\right].$$

Since a positive weighting function does not influence the optimum, we can redefine the weighting function such that the resulting problem takes the stated form:

$$\min_\theta \max_\psi \mathbb{E}_{\substack{\mathbf{y}\sim p_\theta(\mathbf{y}),\\ r\sim\mathcal{U}[0,r_{\max}],\\ \mathbf{x}_r\sim p_r(\mathbf{x}_r|\mathbf{y})}} \lambda(r)\left[\left\|\hat{\mathbf{y}}_\phi^*(\mathbf{x}_r,r)-\mathbf{y}\right\|_2^2-\left\|\hat{\mathbf{y}}_\psi(\mathbf{x}_r,r)-\mathbf{y}\right\|_2^2\right].$$

**Proposition 3.5** (Connection of our IPFM to SiD as $D\to\infty$). *The asymptotic IPFM generator objective (23) and the SiD objective (24) satisfy the following relation:*

$$\mathcal{L}_{IPFM}^{D\to\infty}=2\mathcal{L}_{SiD}-\mathbb{E}_{\substack{\mathbf{y}\sim p_\theta(\mathbf{y})\\ \sigma\sim\mathcal{U}[0,\sigma_{\max}]\\ \mathbf{x}_\sigma\sim p_\sigma(\mathbf{x}_\sigma|\mathbf{y})}}\lambda_\sigma(\sigma)\left\|\hat{\mathbf{y}}_\phi^*(\mathbf{x}_\sigma,\sigma)-\hat{\mathbf{y}}_\psi(\mathbf{x}_\sigma,\sigma)\right\|_2^2.$$

*Proof.* Recall our IPFM asymptotic objective:

$$\mathcal{L}_{IPFM}^{D\to\infty}=\mathbb{E}_{\substack{\mathbf{y}\sim p_\theta(\mathbf{y})\\ \sigma\sim\mathcal{U}[0,\sigma_{\max}]\\ \mathbf{x}_\sigma\sim p_\sigma(\mathbf{x}_\sigma|\mathbf{y})}}\lambda_\sigma(\sigma)\left[\left\|\hat{\mathbf{y}}_\phi^*(\mathbf{x}_\sigma,\sigma)-\mathbf{y}\right\|_2^2-\left\|\hat{\mathbf{y}}_\psi(\mathbf{x}_\sigma,\sigma)-\mathbf{y}\right\|_2^2\right]$$

and the SiD objective:

$$\mathcal{L}_{SiD}=\mathbb{E}_{\substack{\mathbf{y}\sim p_\theta(\mathbf{y})\\ \sigma\sim\mathcal{U}[0,\sigma_{\max}]\\ \mathbf{x}_\sigma\sim p_\sigma(\mathbf{x}_\sigma|\mathbf{y})}}\lambda_\sigma(\sigma)\Big[\left\|\hat{\mathbf{y}}_\phi^*(\mathbf{x}_\sigma,\sigma)-\hat{\mathbf{y}}_\psi(\mathbf{x}_\sigma,\sigma)\right\|_2^2$$
$$+\left\langle\hat{\mathbf{y}}_\phi^*(\mathbf{x}_\sigma,\sigma)-\hat{\mathbf{y}}_\psi(\mathbf{x}_\sigma,\sigma),\hat{\mathbf{y}}_\psi(\mathbf{x}_\sigma,\sigma)-\mathbf{y}\right\rangle\Big].$$

Since the distributions within the expectation and the weighting function $\lambda_\sigma(\sigma)$ are identical for both objectives, it suffices to rewrite our objective's expression under the expectation to reveal the stated connection:

$$\left\|\hat{\mathbf{y}}_\phi^*(\mathbf{x}_\sigma,\sigma)-\mathbf{y}\right\|_2^2-\left\|\hat{\mathbf{y}}_\psi(\mathbf{x}_\sigma,\sigma)-\mathbf{y}\right\|_2^2=$$
$$\|\hat{\mathbf{y}}_\phi^*(\mathbf{x}_\sigma,\sigma)\|_2^2-2\langle\hat{\mathbf{y}}_\phi^*(\mathbf{x}_\sigma,\sigma),\mathbf{y}\rangle+\|\mathbf{y}\|_2^2-\|\hat{\mathbf{y}}_\psi(\mathbf{x}_\sigma,\sigma)\|_2^2+2\langle\hat{\mathbf{y}}_\psi(\mathbf{x}_\sigma,\sigma),\mathbf{y}\rangle-\|\mathbf{y}\|_2^2=$$
$$\|\hat{\mathbf{y}}_\phi^*(\mathbf{x}_\sigma,\sigma)\|_2^2-2\langle\hat{\mathbf{y}}_\phi^*(\mathbf{x}_\sigma,\sigma),\mathbf{y}\rangle-\|\hat{\mathbf{y}}_\psi(\mathbf{x}_\sigma,\sigma)\|_2^2+2\langle\hat{\mathbf{y}}_\psi(\mathbf{x}_\sigma,\sigma),\mathbf{y}\rangle=$$
$$\|\hat{\mathbf{y}}_\phi^*(\mathbf{x}_\sigma,\sigma)\|_2^2-\|\hat{\mathbf{y}}_\psi(\mathbf{x}_\sigma,\sigma)\|_2^2-2\langle\hat{\mathbf{y}}_\phi^*(\mathbf{x}_\sigma,\sigma)-\hat{\mathbf{y}}_\psi(\mathbf{x}_\sigma,\sigma),\mathbf{y}\rangle\pm\|\hat{\mathbf{y}}_\phi^*(\mathbf{x}_\sigma,\sigma)-\hat{\mathbf{y}}_\psi(\mathbf{x}_\sigma,\sigma)\|_2^2=$$
$$\|\hat{\mathbf{y}}_\phi^*(\mathbf{x}_\sigma,\sigma)-\hat{\mathbf{y}}_\psi(\mathbf{x}_\sigma,\sigma)\|_2^2+\|\hat{\mathbf{y}}_\phi^*(\mathbf{x}_\sigma,\sigma)\|_2^2-\|\hat{\mathbf{y}}_\psi(\mathbf{x}_\sigma,\sigma)\|_2^2-2\langle\hat{\mathbf{y}}_\phi^*(\mathbf{x}_\sigma,\sigma)-\hat{\mathbf{y}}_\psi(\mathbf{x}_\sigma,\sigma),\mathbf{y}\rangle-$$
$$\left(\|\hat{\mathbf{y}}_\phi^*(\mathbf{x}_\sigma,\sigma)\|_2^2-2\langle\hat{\mathbf{y}}_\phi^*(\mathbf{x}_\sigma,\sigma),\hat{\mathbf{y}}_\psi(\mathbf{x}_\sigma,\sigma)\rangle+\|\hat{\mathbf{y}}_\psi(\mathbf{x}_\sigma,\sigma)\|_2^2\right)=$$
$$\|\hat{\mathbf{y}}_\phi^*(\mathbf{x}_\sigma,\sigma)-\hat{\mathbf{y}}_\psi(\mathbf{x}_\sigma,\sigma)\|_2^2-\|\hat{\mathbf{y}}_\psi(\mathbf{x}_\sigma,\sigma)\|_2^2-2\langle\hat{\mathbf{y}}_\phi^*(\mathbf{x}_\sigma,\sigma)-\hat{\mathbf{y}}_\psi(\mathbf{x}_\sigma,\sigma),\mathbf{y}\rangle-$$
$$\left(-2\langle\hat{\mathbf{y}}_\phi^*(\mathbf{x}_\sigma,\sigma),\hat{\mathbf{y}}_\psi(\mathbf{x}_\sigma,\sigma)\rangle+\|\hat{\mathbf{y}}_\psi(\mathbf{x}_\sigma,\sigma)\|_2^2\right)=$$
$$\|\hat{\mathbf{y}}_\phi^*(\mathbf{x}_\sigma,\sigma)-\hat{\mathbf{y}}_\psi(\mathbf{x}_\sigma,\sigma)\|_2^2-$$
$$2\left(\langle\hat{\mathbf{y}}_\phi^*(\mathbf{x}_\sigma,\sigma)-\hat{\mathbf{y}}_\psi(\mathbf{x}_\sigma,\sigma),\mathbf{y}\rangle+\langle\hat{\mathbf{y}}_\psi(\mathbf{x}_\sigma,\sigma),\hat{\mathbf{y}}_\psi(\mathbf{x}_\sigma,\sigma)\rangle-\langle\hat{\mathbf{y}}_\phi^*(\mathbf{x}_\sigma,\sigma),\hat{\mathbf{y}}_\psi(\mathbf{x}_\sigma,\sigma)\rangle\right)=$$
$$\|\hat{\mathbf{y}}_\phi^*(\mathbf{x}_\sigma,\sigma)-\hat{\mathbf{y}}_\psi(\mathbf{x}_\sigma,\sigma)\|_2^2-$$
$$2\left(\langle\hat{\mathbf{y}}_\phi^*(\mathbf{x}_\sigma,\sigma)-\hat{\mathbf{y}}_\psi(\mathbf{x}_\sigma,\sigma),\mathbf{y}\rangle+\langle\hat{\mathbf{y}}_\phi^*(\mathbf{x}_\sigma,\sigma)-\hat{\mathbf{y}}_\psi(\mathbf{x}_\sigma,\sigma),\hat{\mathbf{y}}_\psi(\mathbf{x}_\sigma,\sigma)\rangle\right)=$$
$$\|\hat{\mathbf{y}}_\phi^*(\mathbf{x}_\sigma,\sigma)-\hat{\mathbf{y}}_\psi(\mathbf{x}_\sigma,\sigma)\|_2^2-2\langle\hat{\mathbf{y}}_\phi^*(\mathbf{x}_\sigma,\sigma)-\hat{\mathbf{y}}_\psi(\mathbf{x}_\sigma,\sigma),\mathbf{y}-\hat{\mathbf{y}}_\psi(\mathbf{x}_\sigma,\sigma)\rangle=$$
$$\|\hat{\mathbf{y}}_\phi^*(\mathbf{x}_\sigma,\sigma)-\hat{\mathbf{y}}_\psi(\mathbf{x}_\sigma,\sigma)\|_2^2+2\langle\hat{\mathbf{y}}_\phi^*(\mathbf{x}_\sigma,\sigma)-\hat{\mathbf{y}}_\psi(\mathbf{x}_\sigma,\sigma),\hat{\mathbf{y}}_\psi(\mathbf{x}_\sigma,\sigma)-\mathbf{y}\rangle.$$

Therefore, our IPFM asymptotic objective can be rewritten as

$$\mathcal{L}_{IPFM}^{D\to\infty}=\mathbb{E}_{\substack{\mathbf{y}\sim p_\theta(\mathbf{y})\\ \sigma\sim\mathcal{U}[0,\sigma_{\max}]\\ \mathbf{x}_\sigma\sim p_\sigma(\mathbf{x}_\sigma|\mathbf{y})}}\lambda_\sigma(\sigma)\Big[\left\|\hat{\mathbf{y}}_\phi^*(\mathbf{x}_\sigma,\sigma)-\hat{\mathbf{y}}_\psi(\mathbf{x}_\sigma,\sigma)\right\|_2^2$$
$$+2\left\langle\hat{\mathbf{y}}_\phi^*(\mathbf{x}_\sigma,\sigma)-\hat{\mathbf{y}}_\psi(\mathbf{x}_\sigma,\sigma),\hat{\mathbf{y}}_\psi(\mathbf{x}_\sigma,\sigma)-\mathbf{y}\right\rangle\Big],$$

which yields the following relationship with the SiD objective:

$$\mathcal{L}_{\text{IPFM}}^{D\to\infty} = 2\mathcal{L}_{\text{SiD}} - \mathbb{E}_{\substack{\mathbf{y}\sim p_\theta(\mathbf{y})\\ \sigma\sim\mathcal{U}[0,\sigma_{\max}]\\ \mathbf{x}_\sigma\sim p_\sigma(\mathbf{x}_\sigma|\mathbf{y})}} \lambda_\sigma(\sigma)\left\|\hat{\mathbf{y}}_\phi^*(\mathbf{x}_\sigma,\sigma) - \hat{\mathbf{y}}_\psi(\mathbf{x}_\sigma,\sigma)\right\|_2^2.$$

## B EXPERIMENTAL DETAILS

**Hyperparameters.** Our theoretical connection to SiD in §3.2 and §3.3 enables direct transfer of well-tuned hyperparameters from SiD to our method for the diffusion ($D \to \infty$) regime. Following PFGM++ (§2.3), we extend these hyperparameters to finite $D$ settings. We first recall the key hyperparameters from SiD, then describe their adaptation for finite $D$.

*Student model hyperparameters.* For updating the student denoising model $\hat{\mathbf{y}}_\psi(\mathbf{x}_\sigma,\sigma)$, we adopt the noise-level distribution $p(\sigma)$ and weighting function $\lambda_\sigma(\sigma)$ directly from the teacher model's configuration (Karras et al., 2022).

*Generator hyperparameters.* The settings for the generator $G_\theta$ differ. The noise-level distribution $p(\sigma)$ uses parameters $\sigma_{\min} = 0.002$, $\sigma_{\max} = 80$, $\rho = 7$, and $t_{\max} \in [0,1]$. A time step is sampled via $t \sim \text{Unif}[0, t_{\max}]$, with the corresponding noise level given by:

$$\sigma(t) = \left(\sigma_{\max}^{1/\rho} + (1-t)\left(\sigma_{\min}^{1/\rho} - \sigma_{\max}^{1/\rho}\right)\right)^\rho. \tag{43}$$

The weighting function uses the total number of pixels $C$ in an image and the $L_1$ norm with stop-gradient:

$$\lambda_\sigma(\sigma) = C \,/\, \|\hat{\mathbf{y}}_\phi^*(\mathbf{x}_\sigma,\sigma) - \mathbf{y}\|_{1,\text{sg}}. \tag{44}$$

We initialize both the student denoising model $\hat{\mathbf{y}}_\psi$ and generator $G_\theta$ with the pre-trained weights of the teacher model $\hat{\mathbf{y}}_\phi^*$. Following SiD, the generator is applied as $G_\theta(\mathbf{z})$, where $\mathbf{z} \sim p_z = \mathcal{N}(0, \sigma_{\text{init}}^2\mathbf{I})$.

*Adaptation for finite $D$.* Following PFGM++, we adapt the SiD hyperparameters using the reparameterization $r = \sigma\sqrt{D}$:

- Noise-level distribution $p(r)$: Sample $\sigma \sim p(\sigma)$, then set $r = \sigma\sqrt{D}$
- Generator input: Sample from $p_z = p_{r=\sigma_{\text{init}}\sqrt{D}}(\mathbf{x}_r \mid \mathbf{y} = \mathbf{0})$ instead of $\mathcal{N}(0, \sigma_{\text{init}}^2\mathbf{I})$
- Weighting function: Set $\lambda(r) = \lambda_\sigma(\sigma = r/\sqrt{D})$

**SiD-inspired regularization.** The original SiD work (Zhou et al., 2024) found that the regularization introduced in §3.3 improves performance, with larger $\alpha$ values accelerating convergence at the risk of instability. The authors identified $\alpha = 1.0$ and $\alpha = 1.2$ as optimal choices.

Motivated by this finding and our theoretical connection in §3.3, we incorporated SiD-inspired regularization into our framework. However, using $\alpha = 1.2$ with a $D \to \infty$ teacher (i.e., a diffusion model) caused training divergence on CIFAR-10, despite matching SiD's setup. We identified a key difference: our PFGM++ teacher uses the `ncsn++` architecture Karras et al. (2022), whereas SiD employs `ddpm++` Karras et al. (2022). When we repeated the experiment with a `ddpm++` teacher, training stabilized and convergence matched SiD's reported results.

This architecture-dependent sensitivity persisted for finite $D$: with $D = 128$, the `ncsn++` teacher also diverged at $\alpha = 1.2$, while training remained stable for $D = 2048$. Based on this ablation study, we selected $\alpha = 1.0$ for all experiments with SiD-inspired regularization, as it provides a stable and effective balance across all architectures and $D$ values.

**Evaluation.** We assess sample quality using the Fréchet Inception Distance (FID) computed between 50k generated samples and the training set for each dataset. Following established practices in Karras et al. (2022) and PFGM++ (Xu et al., 2023), we perform 3 independent FID evaluations and report the minimum value.

## C  ALGORITHM BOXES

---

**Algorithm 1** Inverse Poisson Flow Matching (IPFM)

---

**Require:** Generator $G_\theta$, dimensionality parameter $D$, $\sigma_{\text{init}} = 2.5$, $\sigma_{\min} = 0.002$,
 1: **Initialization:** $\theta \leftarrow \phi$, $\psi \leftarrow \phi$
 2: **repeat**
 3:     *# Update student denoising model $\hat{\mathbf{y}}_\psi$*
 4:     Sample $R \sim p_{r=\sigma_{\text{init}}\sqrt{D}}(R)$ and $\mathbf{v} = \frac{\mathbf{u}}{\|\mathbf{u}\|}$, $\mathbf{u} \sim \mathcal{N}(0, \mathbf{I})$
 5:     Set $\mathbf{y} = G_\theta(R\mathbf{v})$
 6:     Sample $\sigma \sim p(\sigma)$ (as in Karras et al. (2022))
 7:     Sample $R \sim p_{r=\sigma\sqrt{D}}(R)$ and $\mathbf{v} = \frac{\mathbf{u}}{\|\mathbf{u}\|}$, $\mathbf{u} \sim \mathcal{N}(0, \mathbf{I})$
 8:     Set $\mathbf{x}_r = \mathbf{y} + R\mathbf{v}$
 9:     Update $\psi$ via:

$$\hat{\mathcal{L}}_\psi = \lambda(\sigma) \left\| \hat{\mathbf{y}}_\psi(\mathbf{x}_r, r) - \mathbf{y} \right\|_2^2,$$

$$\psi \leftarrow \psi - \eta \nabla_\psi \hat{\mathcal{L}}_\psi,$$

     where $\lambda(\sigma)$ is defined as in Karras et al. (2022).

10:     *# Update generator $G_\theta$*
11:     Sample $R \sim p_{r=\sigma_{\text{init}}\sqrt{D}}(R)$ and $\mathbf{v} = \frac{\mathbf{u}}{\|\mathbf{u}\|}$, $\mathbf{u} \sim \mathcal{N}(0, \mathbf{I})$
12:     Set $\mathbf{y} = G_\theta(R\mathbf{v})$
13:     Sample $t \sim \mathcal{U}[0, t_{\max}]$ and compute $\sigma$ using (43)
14:     Sample $R \sim p_{r=\sigma\sqrt{D}}(R)$ and $\mathbf{v} = \frac{\mathbf{u}}{\|\mathbf{u}\|}$, $\mathbf{u} \sim \mathcal{N}(0, \mathbf{I})$
15:     Set $\mathbf{x}_r = \mathbf{y} + R\mathbf{v}$
16:     Update $\theta$ via:

$$\hat{\mathcal{L}}_\theta = \lambda(\sigma) \left[ \left\| \hat{\mathbf{y}}_\phi(\mathbf{x}_r, r) - \mathbf{y} \right\|_2^2 - \left\| \hat{\mathbf{y}}_\psi(\mathbf{x}_r, r) - \mathbf{y} \right\|_2^2 \right],$$

$$\theta \leftarrow \theta - \eta \nabla_\theta \hat{\mathcal{L}}_\theta,$$

     where $\lambda(\sigma)$ is defined in (44).
17: **until** FID plateaus or training budget is exhausted
18: **Return** $G_\theta$

---

**Algorithm 2** Multi-step Generator Sampling

---

**Require:** Generator $G_\theta$, dimensionality parameter $D$, number of steps $N$, $\sigma_{\text{init}} = 2.5$, $\sigma_{\min} = 0.02$
 1: Sample $R \sim p_{r=\sigma_{\text{init}}\sqrt{D}}(R)$ and $\mathbf{v} = \frac{\mathbf{u}}{\|\mathbf{u}\|}$, $\mathbf{u} \sim \mathcal{N}(0, \mathbf{I})$
 2: Set $\mathbf{y} = G_\theta(R\mathbf{v})$
 3: **for** $n = 1$ to $N - 1$ **do**
 4:     Set $\sigma_n = \sigma_{\text{init}} + \frac{n-1}{N}(\sigma_{\min} - \sigma_{\text{init}})$
 5:     Sample $R \sim p_{r=\sigma_n\sqrt{D}}(R)$ and $\mathbf{v} = \frac{\mathbf{u}}{\|\mathbf{u}\|}$, $\mathbf{u} \sim \mathcal{N}(0, \mathbf{I})$
 6:     Set $\mathbf{y} = G_\theta(\mathbf{y} + R\mathbf{v})$
 7: **end for**
 8: **Return** $\mathbf{y}$

---

## D  ADDITIONAL EXPERIMENTAL RESULTS

### D.1  COMPARISON TO CONSISTENCY TRAJECTORY MODELS

In this section, we provide a sanity-check experiment demonstrating that Consistency Trajectory Models (Kim et al., 2023, CTM), an advanced general ODE-based distillation method, is surpassed by our IPFM when distilling a finite-$D$ PFGM++ teacher without GAN-loss. Note that for the infinite $D$ case, the superior performance of our IPFM stems from two facts: 1) our IPFM reduces

to SiD in the diffusion ($D \rightarrow \infty$) limit (see Section 3.3), and 2) SiD (Zhou et al., 2024, Section 5) surpasses CTM.

For this experiment, we use a PFGM++ ($D = 128$) teacher trained on CIFAR-10. We intentionally omit adding GAN-losses for both methods to compare their fundamental distillation performance.

To evaluate CTM distillation, we use the official repository[1] specifically prepared for the CIFAR-10 task. To adapt this repository for the finite-$D$ case and ensure fair comparison, we make the following modifications: 1) we adapt the code for the finite-$D$ perturbation kernel following the approach in PFCM (Hein et al., 2025), and 2) we disable mixed precision calculations and use only full precision, as this should not degrade results and matches our IPFM setup. Other hyperparameters remain unchanged.

We train CTM for a hundred thousand update steps, which takes more hours than our single-step distillation. To align the maximum time values, we slightly extend the training of our IPFM ($D = 128$, $\alpha = 1.0$) from the main experiments.

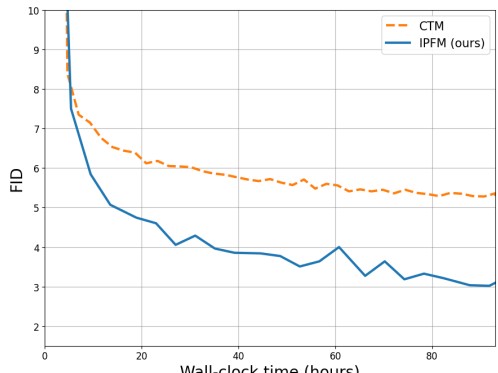

Figure 3: A convergence comparison between single-step generators produced by CTM and our IPFM.

Convergence plots in Figure 3 show that the general ODE-based CTM converges significantly slower than our IPFM. While CTM maintains FID over 5 throughout the entire training period, our IPFM achieves FID of 3 within the same timeframe. Moreover, the CTM curve appears close to a plateau, while our IPFM continues to show a steady reduction in FID.

## D.2  CLASS-CONDITIONAL CIFAR-10

In this section, we provide additional experiments on class-conditional CIFAR-10. The results are consistent with those for unconditional CIFAR-10 and FFHQ 64x64. Specifically, Table 2 shows that our unregularized IPFM matches the teacher's quality in just a few steps, while the regularized version can even outperform it. Furthermore, Figure 4 shows that one-step generators with finite $D$ exhibit faster early convergence, although with a smaller margin than in the unconditional setting.

Table 2: **Quantitative results of our IPFM distillation on class-conditional CIFAR-10.** We report FID scores for distilled generators across different auxiliary dimensions $D$, number of function evaluations (NFE), and regularization strengths $\alpha$. Note that (a) $\alpha = 0.5$ corresponds to the unregularized IPFM and (b) $D \rightarrow \infty$ with $\alpha = 1.0$ recovers SiD (the corresponding rows are highlighted). For comparison, we include the original PFGM++ teacher models evaluated in few-step regimes, demonstrating that distillation is essential for high-quality few-step generation. In each row (corresponding to a specific $D$), the best FID score is **bolded** and the second best is underlined.

| **D** | **α** | **IPFM** (ours) | | | **Teacher** (PFGM++, class-conditional CIFAR-10) | | | | | |
|---|---|---|---|---|---|---|---|---|---|---|
| | | 1 | 2 | 4 | 1 | 5 | 9 | 17 | 25 | 35 |
| 2048 | 0.5 | 4.43 | 2.54 | 1.81 | >100 | >100 | 35.59 | 3.13 | 1.90 | 1.74 |
| | 1.0 | 3.13 | 1.96 | **1.64** | | | | | | |
| ∞ (Diffusion) | 0.5 | 4.36 | 2.37 | 1.86 | >100 | >100 | 34.88 | 3.21 | 1.94 | 1.81 |
| | 1.0 | 3.16 | 1.97 | **1.64** | | | | | | |

---

[1]https://github.com/Kim-Dongjun/ctm-cifar10

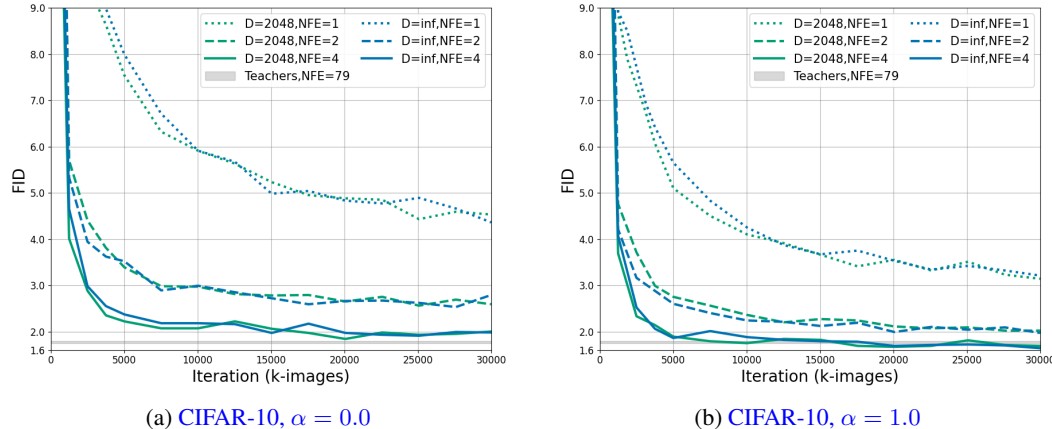

(a) CIFAR-10, $\alpha = 0.0$          (b) CIFAR-10, $\alpha = 1.0$

Figure 4: **Convergence of our IPFM on class-conditional CIFAR-10.** Plots show FID evolution during training (measured in thousands of generator samples) for different IPFM settings. The gray horizontal band represents the range of final performance (minimum to maximum FID across different D) achieved by the original PFGM++ teacher models, providing a target quality threshold.

### D.3 ABLATION OF THE REGULARIZATION STRENGTH $\alpha$

This section presents an ablation study on the regularization strength parameter $\alpha$ for finite-$D$ models. We trained our IPFM using a PFGM++ ($D = 128$) teacher on the FFHQ 64x64 dataset, evaluating values of $\alpha \in \{0.0, 0.5, 0.6, 0.8, 1.0\}$ during 15,000 generated images.

As shown in Figure 5, stronger regularization (higher $\alpha$ values) yields faster convergence and lower final FID. This finding aligns with observations in SiD (Zhou et al., 2024, Appendix A), suggesting that the benefits of stronger regularization during distillation extend from diffusion models to finite-$D$ electrostatic models.

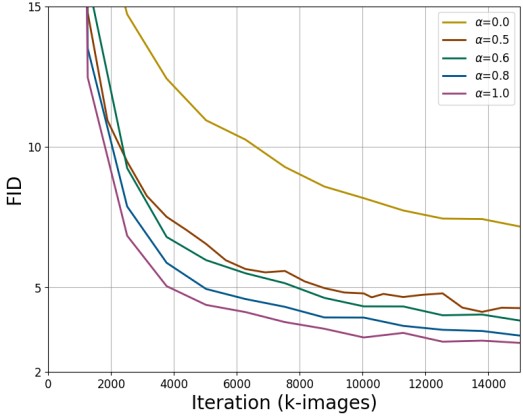

Figure 5: Ablation of regularization strength $\alpha$ on FFHQ for a finite-$D$ PFGM++ teacher ($D = 128$).

### D.4 ADDITIONAL ANALYSIS OF NFE PERFORMANCE OF TEACHER MODELS

This section verifies that the Number of Function Evaluations (NFE) used for teacher models in our main experiments represent their full generative capability, thereby validating our claims of matching or surpassing teacher performance. We evaluated the teachers at higher NFE values to confirm that their performance had saturated at the NFEs used in our primary analysis. As shown in

Table 3, FID scores remain stable even at substantially higher NFEs (up to 999), indicating that the teachers had indeed converged at our chosen evaluation points.

Note that for consistency across all NFE values in this analysis, we computed FID a single time using 50,000 generated images. This slightly differs from the evaluation protocol used in our main experiments, which employed multiple random seeds as detailed in Appendix B.

Table 3: **Teacher Model Performance at Higher NFE Values.**

| D | Our Best | Teacher (PFGM++, CIFAR-10) | | | |
|---|---|---|---|---|---|
| | IPFM | 35 | 99 | 199 | 999 |
| 128 | **1.75** | 1.92*/1.92 | 1.94 | 1.97 | 1.97 |
| 2048 | **1.82** | 1.91*/1.91 | 1.95 | 1.95 | 1.97 |
| ∞ | **1.86** | 1.98*/1.99 | 1.99 | 2.01 | 2.02 |

| D | Our Best | Teacher (PFGM++, FFHQ 64x64) | | | |
|---|---|---|---|---|---|
| | IPFM | 79 | 99 | 199 | 999 |
| 128 | **1.72** | 2.43*/2.47 | 2.46 | 2.45 | 2.44 |
| ∞ | **1.70** | 2.53*/2.53 | 2.53 | 2.52 | 2.52 |

# E  QUALITATIVE RESULTS

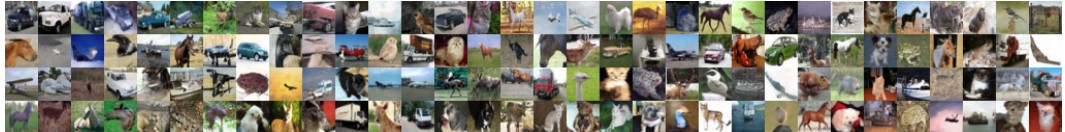

Figure 6: Samples generated with our IPFM (D=128,NFE=1,$\alpha$=0.0) on CIFAR-10 (FID=5.38)

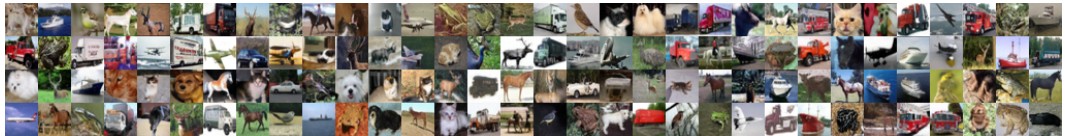

Figure 7: Samples generated with our IPFM (D=128,NFE=2,$\alpha$=0.0) on CIFAR-10 (FID=2.68)

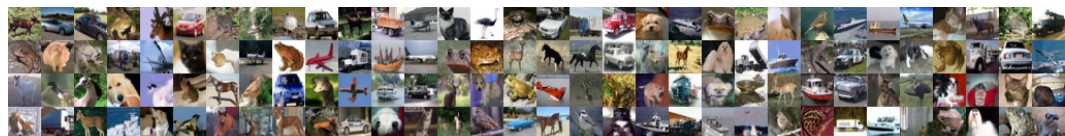

Figure 8: Samples generated with our IPFM (D=128,NFE=4,$\alpha$=0.0) on CIFAR-10 (FID=2.08)

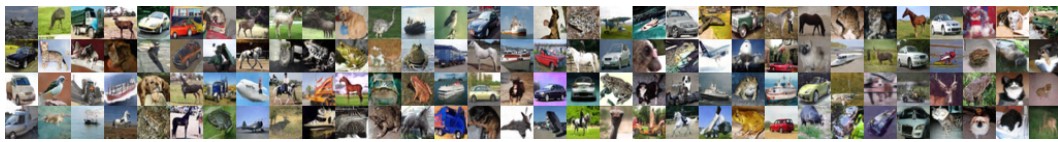

Figure 9: Samples generated with our IPFM (D=128,NFE=1,$\alpha$=1.0) on CIFAR-10 (FID=3.31)

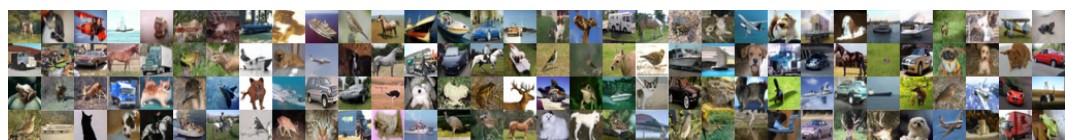

Figure 10: Samples generated with our IPFM (D=128,NFE=2,$\alpha$=1.0) on CIFAR-10 (FID=2.12)

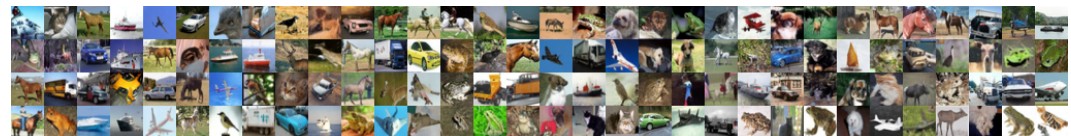

Figure 11: Samples generated with our IPFM (D=128,NFE=4,$\alpha$=1.0) on CIFAR-10 (FID=1.75)

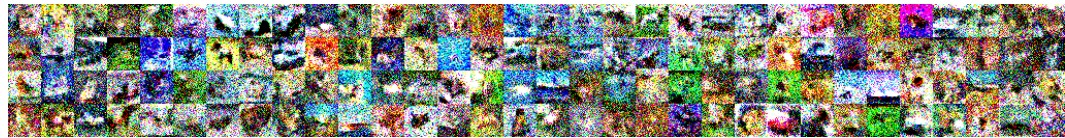

Figure 12: Samples generated with PFGM++ (D=128,NFE=5) teacher on CIFAR-10 (FID>100)

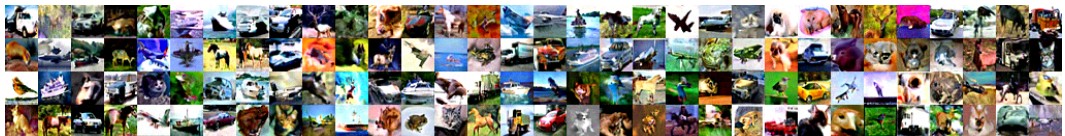

Figure 13: Samples generated with PFGM++ (D=128,NFE=9) teacher on CIFAR-10 (FID=37.79)

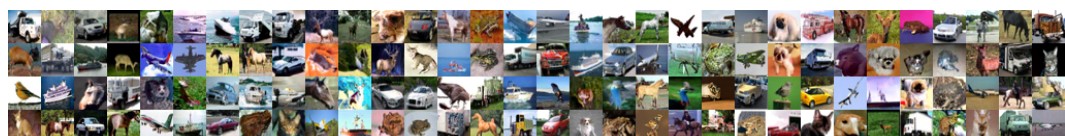

Figure 14: Samples generated with PFGM++ (D=128,NFE=17) teacher on CIFAR-10 (FID=3.32)

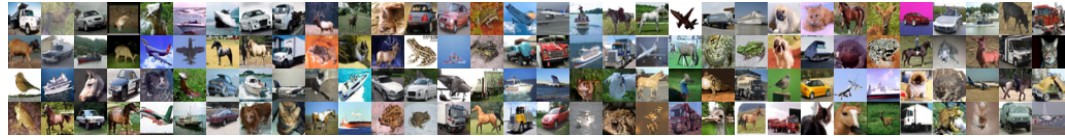

Figure 15: Samples generated with PFGM++ (D=128,NFE=35) teacher on CIFAR-10 (FID=1.92)

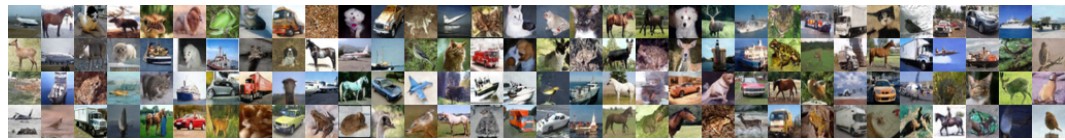

Figure 16: Samples generated with our IPFM (D=2048,NFE=1,$\alpha$=0.0) on CIFAR-10 (FID=5.47)

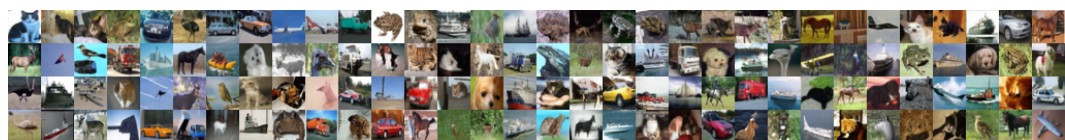

Figure 17: Samples generated with our IPFM (D=2048,NFE=2,$\alpha$=0.0) on CIFAR-10 (FID=3.05)

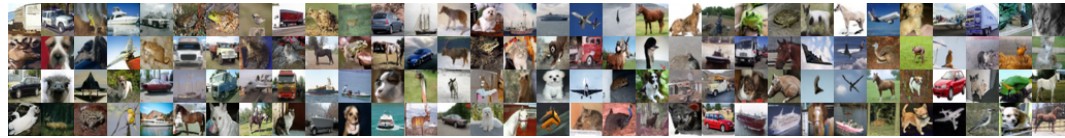

Figure 18: Samples generated with our IPFM (D=2048,NFE=4,$\alpha$=0.0) on CIFAR-10 (FID=2.02)

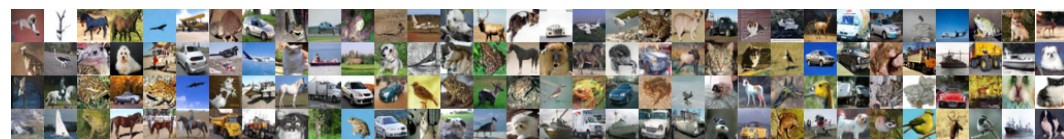

Figure 19: Samples generated with our IPFM (D=2048,NFE=1,$\alpha$=1.0) on CIFAR-10 (FID=3.20)

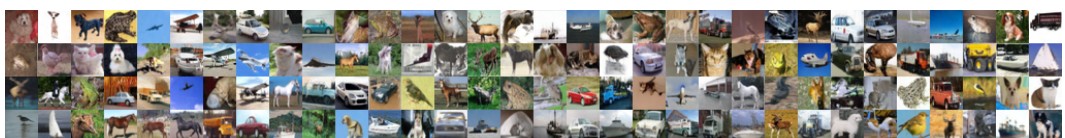

Figure 20: Samples generated with our IPFM (D=2048,NFE=2,$\alpha$=1.0) on CIFAR-10 (FID=2.15)

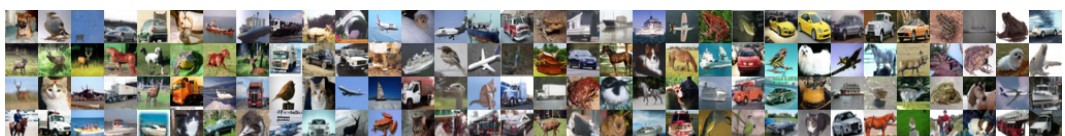

Figure 21: Samples generated with our IPFM (D=2048,NFE=4,$\alpha$=1.0) on CIFAR-10 (FID=1.82)

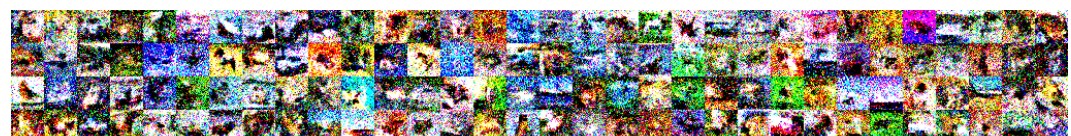

Figure 22: Samples generated with PFGM++ (D=2048,NFE=5) teacher on CIFAR-10 (FID > 100)

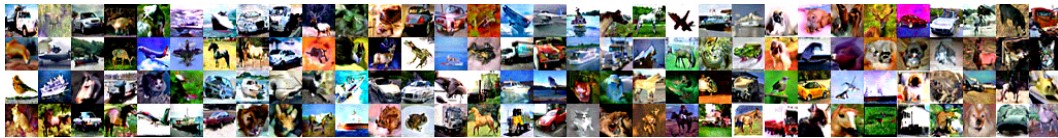

Figure 23: Samples generated with PFGM++ (D=2048,NFE=9) teacher on CIFAR-10 (FID=37.14)

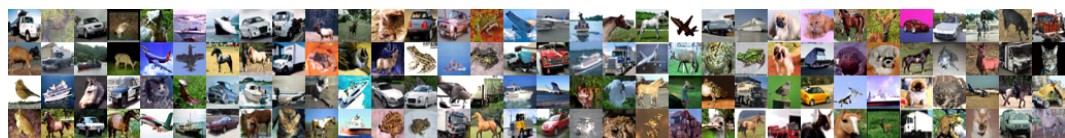

Figure 24: Samples generated with PFGM++ (D=2048,NFE=17) teacher on CIFAR-10 (FID=3.37)

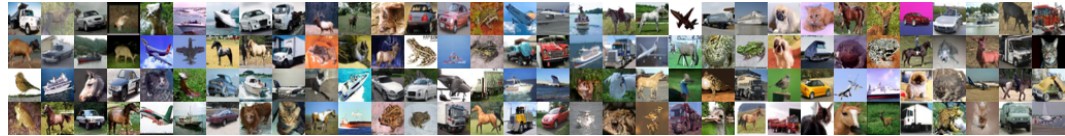

Figure 25: Samples generated with PFGM++ (D=2048,NFE=35) teacher on CIFAR-10 (FID=1.91)

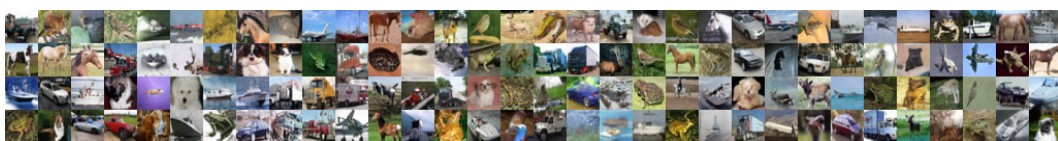

Figure 26: Samples generated with our IPFM (D=inf,NFE=1,$\alpha$=0.0) on CIFAR-10 (FID=5.57)

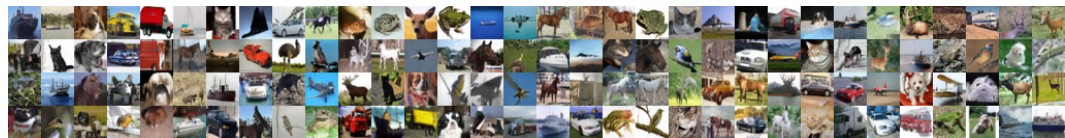

Figure 27: Samples generated with our IPFM (D=inf,NFE=2,$\alpha$=0.0) on CIFAR-10 (FID=2.86)

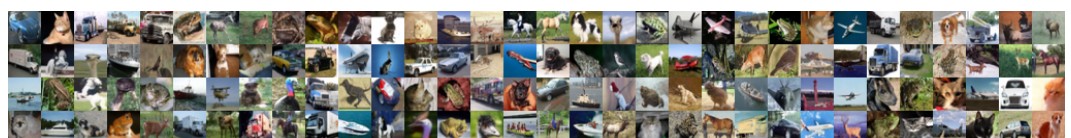

Figure 28: Samples generated with our IPFM (D=inf,NFE=4,$\alpha$=0.0) on CIFAR-10 (FID=2.13)

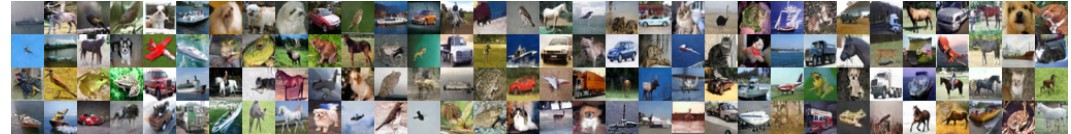

Figure 29: Samples generated with our IPFM (D=inf,NFE=1,$\alpha$=1.0) on CIFAR-10 (FID=3.47)

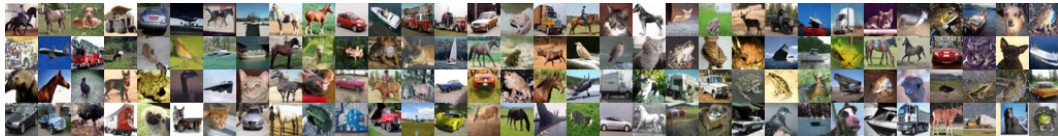

Figure 30: Samples generated with our IPFM (D=inf,NFE=2,$\alpha$=1.0) on CIFAR-10 (FID=2.15)

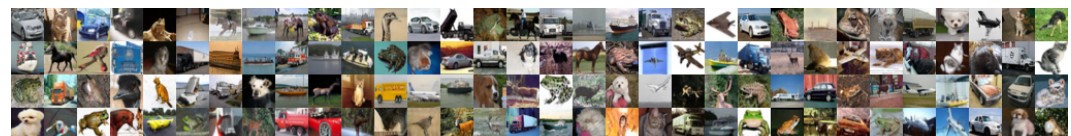

Figure 31: Samples generated with our IPFM (D=inf,NFE=4,$\alpha$=1.0) on CIFAR-10 (FID=1.86)

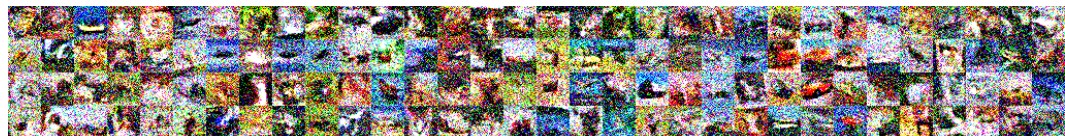

Figure 32: Samples generated with PFGM++ (D=inf,NFE=5) teacher on CIFAR-10 (FID>100)

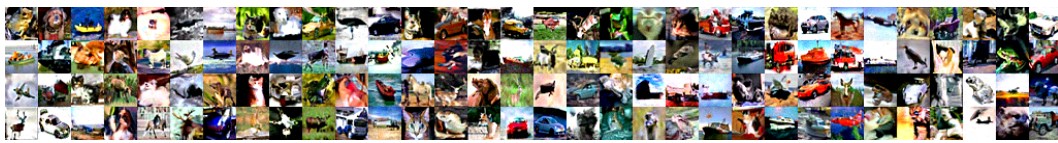

Figure 33: Samples generated with PFGM++ (D=inf,NFE=9) teacher on CIFAR-10 (FID=40.24)

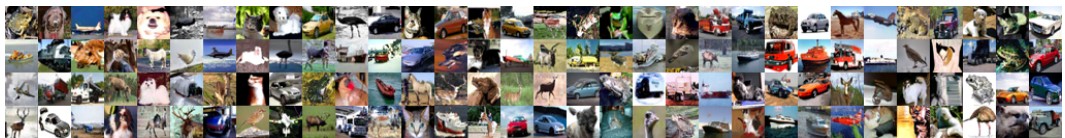

Figure 34: Samples generated with PFGM++ (D=inf,NFE=17) teacher on CIFAR-10 (FID=3.74)

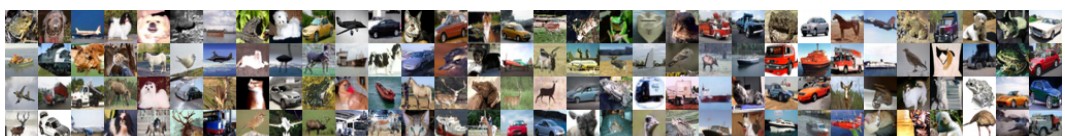

Figure 35: Samples generated with PFGM++ (D=inf,NFE=35) teacher on CIFAR-10 (FID=1.98)

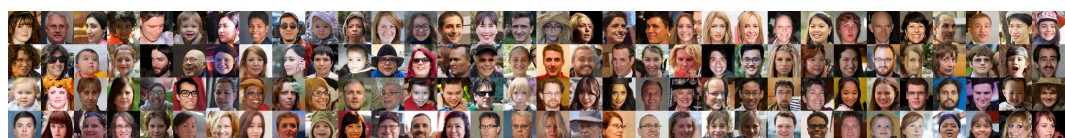

Figure 36: Samples generated with our IPFM (D=128,NFE=1,$\alpha$=0.0) on FFHQ 64x64 (FID=3.42)

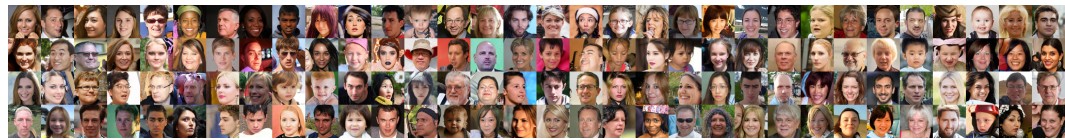

Figure 37: Samples generated with our IPFM (D=128,NFE=2,$\alpha$=0.0) on FFHQ 64x64 (FID=2.12)

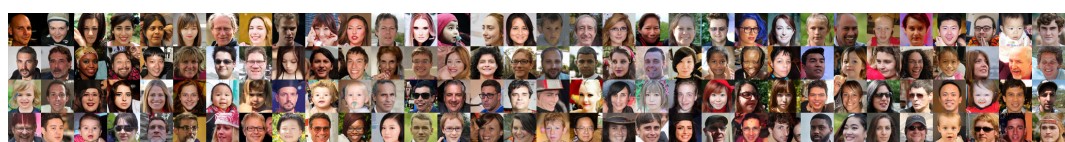

Figure 38: Samples generated with our IPFM (D=128,NFE=1,$\alpha$=1.0) on FFHQ 64x64 (FID=2.40)

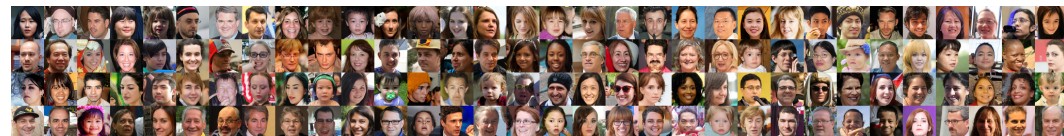

Figure 39: Samples generated with our IPFM (D=128,NFE=2,$\alpha$=1.0) on FFHQ 64x64 (FID=1.72)

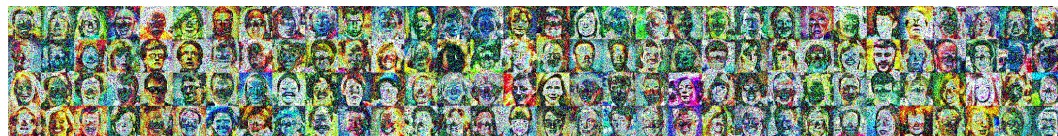

Figure 40: Samples generated with PFGM++ (D=128,NFE=5) teacher on FFHQ 64x64 (FID>100)

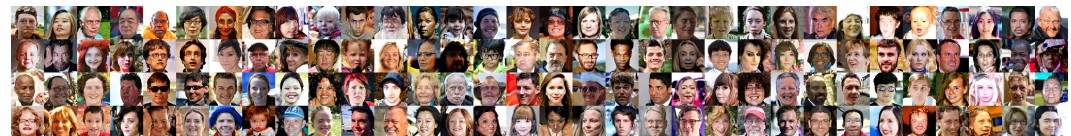

Figure 41: Samples generated with PFGM++ (D=128,NFE=13) teacher on FFHQ 64x64 (FID=16.35)

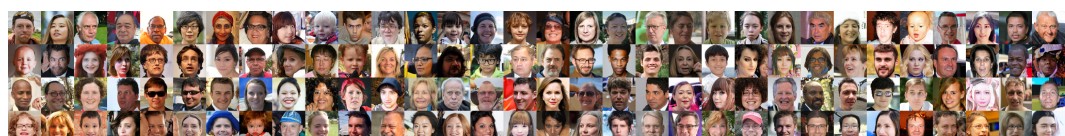

Figure 42: Samples generated with PFGM++ (D=128,NFE=31) teacher on FFHQ 64x64 (FID=2.89)

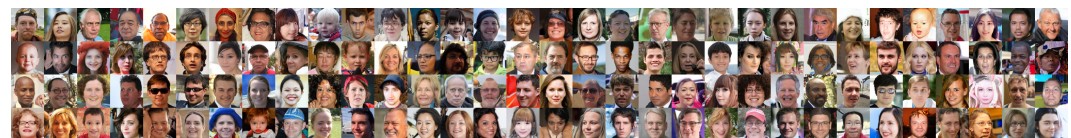

Figure 43: Samples generated with PFGM++ (D=128,NFE=79) teacher on FFHQ 64x64 (FID=2.43)

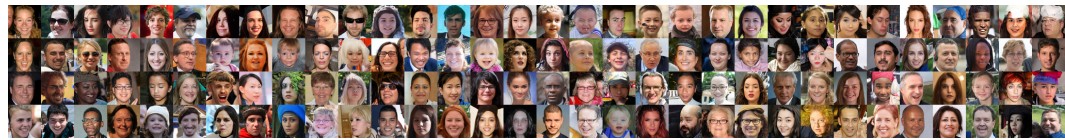

Figure 44: Samples generated with our IPFM (D=inf,NFE=1,$\alpha$=0.0) on FFHQ 64x64 (FID=3.91)

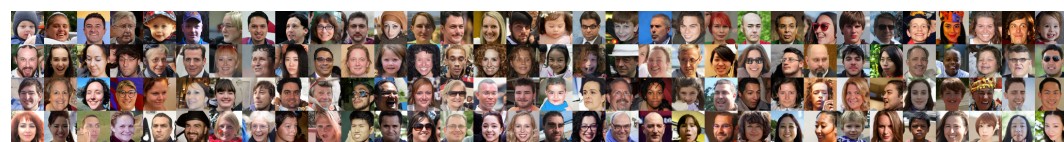

Figure 45: Samples generated with our IPFM (D=inf,NFE=2,$\alpha$=0.0) on FFHQ 64x64 (FID=2.04)

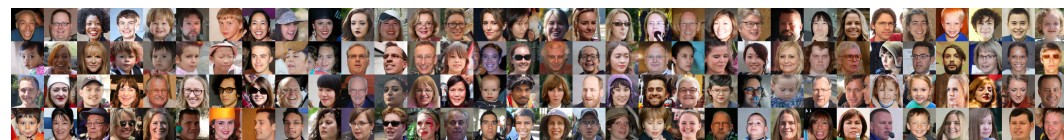

Figure 46: Samples generated with our IPFM (D=inf,NFE=1,$\alpha$=1.0) on FFHQ 64x64 (FID=2.60)

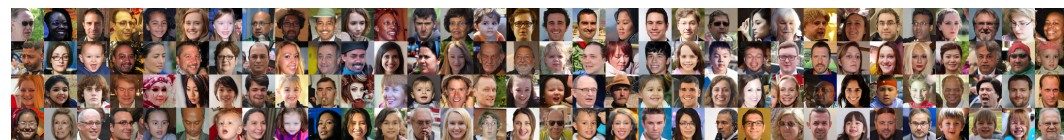

Figure 47: Samples generated with our IPFM (D=inf,NFE=2,$\alpha$=1.0) on FFHQ 64x64 (FID=1.70)

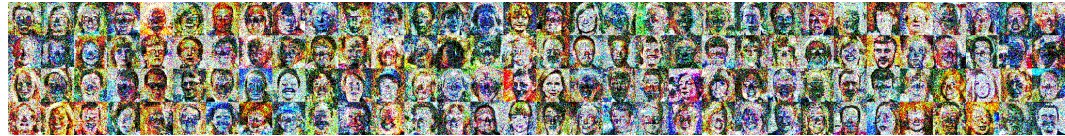

Figure 48: Samples generated with PFGM++ (D=inf,NFE=5) teacher on FFHQ 64x64 (FID>100)

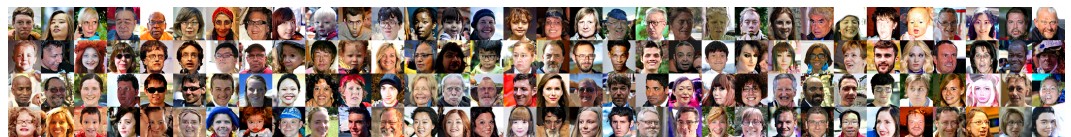

Figure 49: Samples generated with PFGM++ (D=inf,NFE=13) teacher on FFHQ 64x64 (FID=15.82)

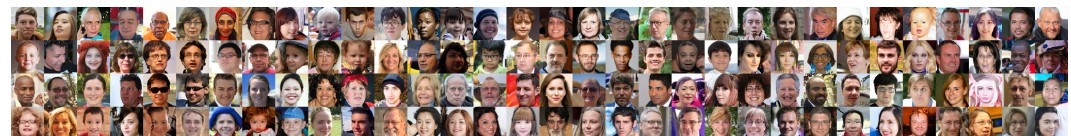

Figure 50: Samples generated with PFGM++ (D=inf,NFE=31) teacher on FFHQ 64x64 (FID=2.84)

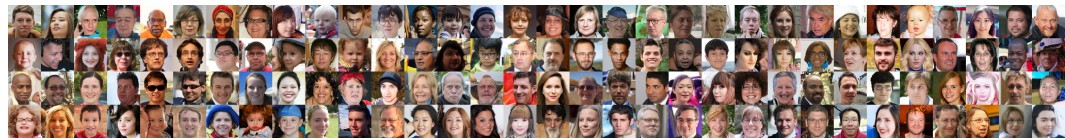

Figure 51: Samples generated with PFGM++ (D=inf,NFE=79) teacher on FFHQ 64x64 (FID=2.53)

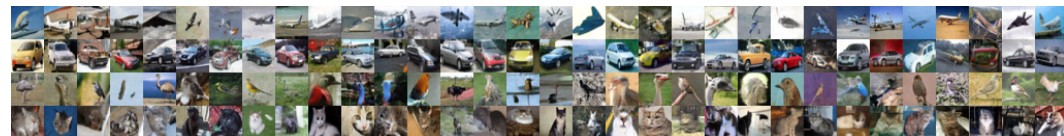

Figure 52: Samples generated with our IPFM (D=2048,NFE=1,$\alpha$=0.0) on class-conditional CIFAR-10 (FID=4.43)

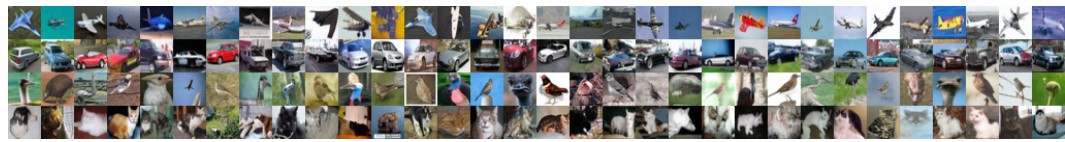

Figure 53: Samples generated with our IPFM (D=2048,NFE=2,$\alpha$=0.0) on class-conditional CIFAR-10 (FID=2.54)

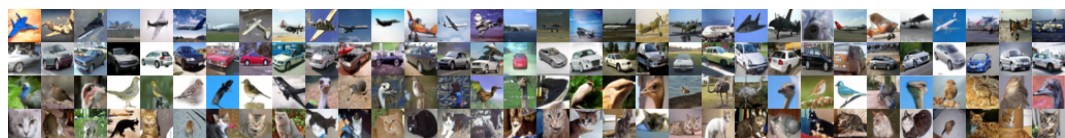

Figure 54: Samples generated with our IPFM (D=2048,NFE=4,$\alpha$=0.0) on class-conditional CIFAR-10 (FID=1.81)

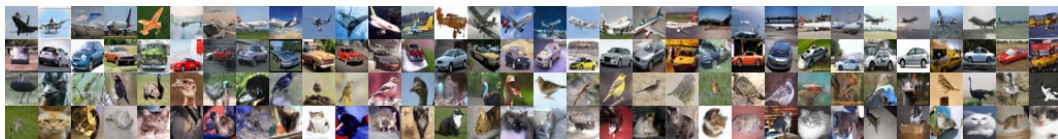

Figure 55: Samples generated with our IPFM (D=2048,NFE=1,$\alpha$=1.0) on class-conditional CIFAR-10 (FID=3.13)

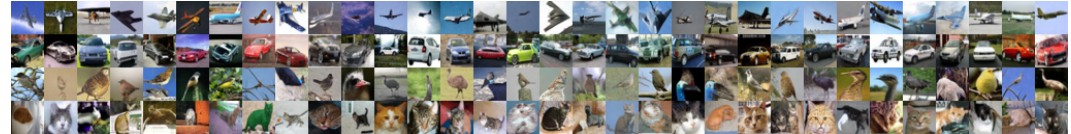

Figure 56: Samples generated with our IPFM (D=2048,NFE=2,$\alpha$=1.0) on class-conditional CIFAR-10 (FID=1.96)

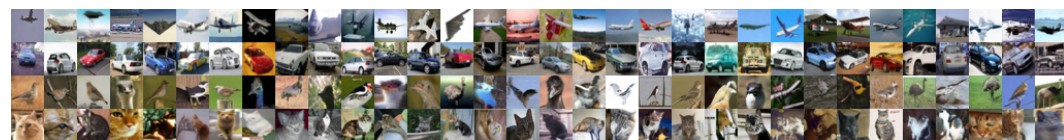

Figure 57: Samples generated with our IPFM (D=2048,NFE=4,$\alpha$=1.0) on class-conditional CIFAR-10 (FID=1.64)

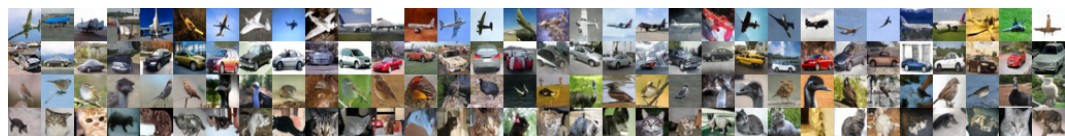

Figure 58: Samples generated with our IPFM (D=inf,NFE=1,$\alpha$=0.0) on class-conditional CIFAR-10 (FID=4.36)

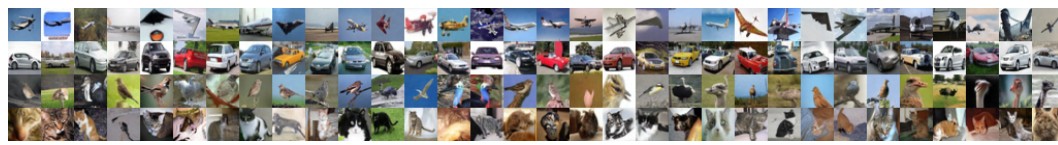

Figure 59: Samples generated with our IPFM (D=inf,NFE=2,$\alpha$=0.0) on class-conditional CIFAR-10 (FID=2.37)

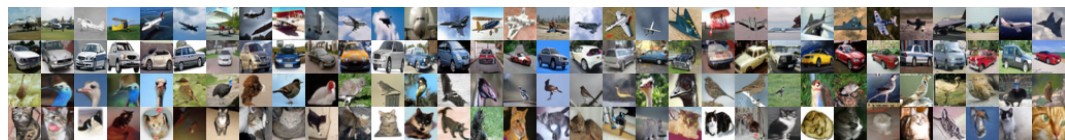

Figure 60: Samples generated with our IPFM (D=inf,NFE=4,$\alpha$=0.0) on class-conditional CIFAR-10 (FID=1.86)

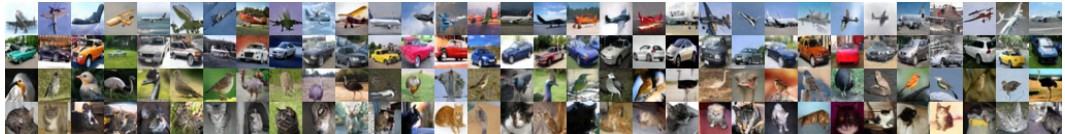

Figure 61: Samples generated with our IPFM (D=inf,NFE=1,$\alpha$=1.0) on class-conditional CIFAR-10 (FID=3.16)

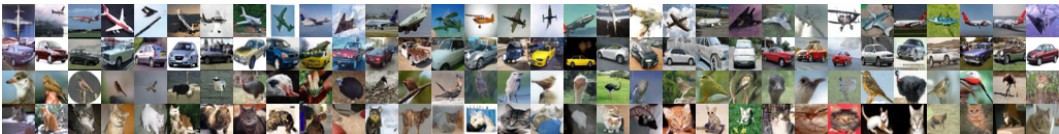

Figure 62: Samples generated with our IPFM (D=inf,NFE=2,$\alpha$=1.0) on class-conditional CIFAR-10 (FID=1.76)

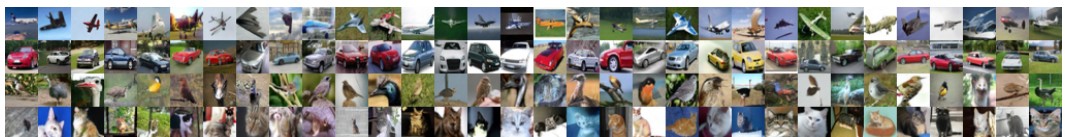

Figure 63: Samples generated with our IPFM (D=inf,NFE=4,$\alpha$=1.0) on class-conditional CIFAR-10 (FID=1.64)

## F  LARGE LANGUAGE MODELS USAGE

Large language models were used solely for grammatical correction and improving text clarity.

