# OpenReview forum: "Overclocking Electrostatic Generative Models"
_ICLR.cc/2026/Conference — Submitted to ICLR 2026_

### Official Review · Reviewer_1SzS · 2025-10-30

**Soundness:** 3
**Presentation:** 3
**Contribution:** 3
**Rating:** 6
**Confidence:** 4

**Summary:**

This paper introduces Inverse Poisson Flow Matching (IPFM), a distillation framework designed to train a single-step generator from a teacher Poisson Flow Generative Model. The method optimizes a generator such that the electrostatic field induced by its output distribution matches that of the real data, a concept closely tied to variational formulations of diffusion distillation. By analyzing the infinite-dimensional limit of Poisson flow, the authors establish a theoretical connection to Score Identity Distillation (SiD) [1], a known diffusion distillation approach. This insight motivates a regularization strategy that further enhances performance. Experimental results on CIFAR-10 and FFHQ demonstrate that IPFM can effectively distill a single-step generator from a teacher model, achieving competitive results.

**Strengths:**

1. **Sound Theoretical Formulation**
   The proposed idea of aligning the electrostatic field of the generator’s output distribution with the ground-truth field is simple and intuitive.

2. **Novel Connection to Diffusion Distillation**
   The connection between Poisson flow matching and SiD [1] is both original and insightful, providing a bridge between finite- and infinite-dimensional flow formulations.

3. **Clear Writing and Presentation**
   The paper is clear and well written

**Weaknesses:**

1. **Lack of Theoretical Guarantees for the Objective**
   While the introduction of the inverse Poisson flow matching loss is intuitive, the paper lacks formal proofs showing that minimizing the loss reduces the divergence of the generator distribution to the target distribution. Additionally, a proof demonstrating that the divergence becomes zero when the loss is zero would strengthen the theoretical soundness and rule out degenerate solutions.

2. **Limited Experimental Evaluation**
   The experiments are restricted to relatively simple datasets such as CIFAR-10 and FFHQ. To validate IPFM’s scalability and robustness, results on more challenging benchmarks, such as **ImageNet**, are needed, particularly since these datasets are common in diffusion and distillation research.

3. **Insufficient Ablation Studies**
   Ablation experiments are limited to only two values of the regularization coefficient ($\alpha$). A broader range of values should be explored, especially the critical case ($\alpha = 1/2$), where the method reduces to the original IPFM without the new regularizer. This would help clarify the contribution of the proposed regularization term.

**Questions:**

1. **Provide Theoretical Proofs for the Objective Function**
   Formal theoretical analysis would help establish guarantees on distributional divergence and justify the use of the proposed objective. Specifically, proofs bounding the divergence between the learned and target distributions, and demonstrating that the divergence is zero when the loss is zero, would improve the paper’s rigor.

2. **Expand Experimental Scope and Ablations**
   Future versions of this work should include larger-scale experiments (e.g., ImageNet) and a more thorough study of the impact of the regularization parameter ($\alpha$). This would validate the method’s generalizability and robustness.

3. **Clarify the Advantage of Finite-Dimensional Poisson Flow Distillation**
   The paper should better articulate the practical advantage of distilling finite-dimensional Poisson flows compared to their infinite-dimensional counterparts. From the reported results, both seem to yield similar outcomes, so further discussion is needed to justify why the finite-dimensional distillation is valuable in practice.

4. **Comparison with Naïve Distillation Baselines**
   Since Poisson flows can also be represented as ODE systems, it would be beneficial to compare IPFM with straightforward trajectory distillation methods such as **Consistency Training Models (CTM)** [2] and **Consistency Models (CM)** [3]. Evaluating these baselines would contextualize the improvements introduced by IPFM and demonstrate its advantages beyond standard ODE-based distillation.

---

## References
[1] **Score identity Distillation: Exponentially Fast Distillation of Pretrained Diffusion Models for One-Step Generation.**
[2] **Consistency Trajectory Models: Learning Probability Flow ODE Trajectory of Diffusion.**
[3] **Consistency Models.**

---

> ### Author Response · Authors · 2025-11-20
> **Answer to Reviewer 1SzS**
>
> **Question 1 + Weakness 1:** Provide a Theoretical Proof for IPFM Well-Posedness.
>
> **Answer:**
>
> Thank you for your remark. We agree that this proof is crucial as it provides theoretical guarantees for our IPFM. **We have updated the manuscript** with the corresponding theorem following the IPFM problem statement and provided the complete proof in Appendix A.
>
> ---
>
> **Question 2 + Weakness 2 + Weakness 3:** Expand Experimental Scope and Ablations.
>
> **Answer:**
>
> **Expanded Scope:** While ImageNet-scale experiments would be valuable, there are no pre-trained PFGM++ teachers available for finite-$D$ at this scale. Since training a finite-$D$ teacher followed by distillation remains beyond our current computational resources, we have strengthened our evaluation by adding **class-conditional generation on CIFAR-10**. This new experiment, reported in Appendix D, uses the corresponding PFGM++ teacher and shows results consistent with our unconditional findings. With this addition, our study now covers **all teacher models released by the PFGM++ authors**. **We have updated the manuscript** with these results in Appendix D.
>
> **Correction to $\alpha$ Values:** We have corrected an important typo in Table 1. The cases previously marked as $\alpha=0$ were intended to represent our unregularized IPFM objective; however, $\alpha=0.5$ is the theoretically correct value for this objective. **We have updated** all instances of $\alpha=0.0$ to $\alpha=0.5$ throughout **the manuscript** and added clearer emphasis on the unregularized IPFM evaluation.
>
> **Ablation of $\alpha$:** We conducted an ablation study of $\alpha \in (0.0, 0.5, 0.6, 0.8, 1.0)$ on the FFHQ dataset using a PFGM++ D=128 teacher. The results demonstrate that higher $\alpha$ values lead to better convergence, which aligns with observations in SiD. **We have included these results in Appendix D of the revised manuscript.**
>
> ---
>
> **Question 3:** Clarify the Advantage of Finite-Dimensional Poisson Flow Distillation.
>
> **Answer:**
>
> You raise an important question regarding the advantages of finite-$D$ models, which is fundamental to the value of electrostatic generative models as an alternative to classical diffusion. If finite-$D$ models are beneficial, then developing distillation techniques for them is inherently valuable.
>
> The PFGM++ framework has already demonstrated that finite-$D$ models possess greater robustness and stability during inference, attributable to the heavier tails of their perturbation kernels. Therefore, our primary goal was to provide a distillation technique to accelerate sampling for these novel and advantageous models, which we have successfully accomplished.
>
> Furthermore, in our work we observed a specific advantage in the distillation process itself: one-step generators for finite-$D$ settings converge faster than their infinite-$D$ (diffusion) counterparts. We hypothesize this is also due to the teacher's robustness and the heavier-tailed kernels. We discuss this finding in the Experiments section under "The Impact of Auxiliary Dimension D".
>
> ---
>
> **Question 4:** Comparison with Naïve Distillation Baselines.
>
> **Answer:**
>
> To the best of our knowledge, SiD is the only GAN-free distillation method that can match (or even surpass) the performance of a diffusion teacher (the $D \to \infty$ regime of our IPFM). Our results show that finite-$D$ variants of our IPFM converge at least as well as, and in some cases better than, this diffusion limit. This indicates that our IPFM can attain teacher-level performance in a single step given sufficient compute.  By contrast, other GAN-free distillation approaches (e.g., consistency-based models) have not been demonstrated to reach teacher performance even for diffusion teachers. Moreover, there are no specific adaptations of these approaches for PFGM++ teachers on standard generative benchmarks. For this reason, we do not include other baselines for finite-$D$ Poisson-flow distillation and focus our evaluation on the theoretically grounded connection between our IPFM and SiD across different values of $D$. **We have updated the revised manuscript** with this discussion in the Discussion section.
>
> ---
>
> **Conclusion remarks:**
>
> We hope these clarifications have fully addressed your concerns. Should any points require further elaboration, we are happy to provide it. Based on these comprehensive explanations, we are hopeful that you will consider raising your rating.

---

> > ### Comment · Reviewer_1SzS · 2025-11-24
> >
> > I thank the authors for taking the time to respond and appreciate the effort to provide a proof of the validity of the IPFM loss, but several concerns remain. With respect to Figure 2, I do not clearly see faster convergence of the finite-dimensional method compared to the infinite-dimensional one for NFE = 2 and 4, and the figure only reflects mid-training behavior; showing full training curves would help clarify whether the claimed convergence advantage persists up to convergence. The presentation of the proof could also be improved: the use of informal notation such as “const” instead of explicitly defined constants, and the organization of the argument as a sequence of steps rather than a more structured development via propositions or lemmas leading to a main theorem, make the proof harder to follow and less polished. Finally, although I agree that SiD has been shown in prior work to outperform trajectory-based distillation techniques, I still believe that presenting experimental results in this paper would substantially strengthen the case for introducing a new distillation method for Poisson flows. Overall, I remain positive about this work, but given these remaining concerns I prefer to keep my original score.

---

> ### Author Response · Authors · 2025-12-03
> **Answer to Reviewer 1SzS**
>
> **Comment 1:** Finally, although I agree that SiD has been shown in prior work to outperform trajectory-based distillation techniques, I still believe that presenting experimental results in this paper would substantially strengthen the case for introducing a new distillation method for Poisson flows.
>
> **Answer:**
>
> We agree that showing superior performance over consistency-based methods more explicitly would further strengthen our work. To this end, we added a sanity-check experiment comparing our IPFM to Consistency Trajectory Models, an advanced consistency-based distillation, in finite-$D$ setting. The results show the superior performance of our IPFM, highlighting the importance of our method. **We have updated the revised manuscript** with this experiment in the Appendix D and added relevant comments in the Introduction and Discussion sections.
>
> ---
>
> **Comment 2:** The presentation of the proof could also be improved: the use of informal notation such as “const” instead of explicitly defined constants, and the organization of the argument as a sequence of steps rather than a more structured development via propositions or lemmas leading to a main theorem, make the proof harder to follow and less polished.
>
> **Answer:**
>
> We have revised the manuscript to improve the clarity and structure of the proof.
>
> First, **we have updated the notation** by replacing the informal "$const$" with more conventional "$c$", to eliminate ambiguity. We would also like to reiterate that, as was stated for the original notation, this constant "$c$" does not depend on $\mathbf{x}_r$.
>
> Second, **we have improved the proof's structure**. We have added a concluding summary that explicitly synthesizes the steps to show how they collectively prove the theorem. Furthermore, we now precede each major step with the word "Proof." to indicate that it can be considered an independent, lemma-like argument.
>
> ---
>
> **Comment 3:** With respect to Figure 2, I do not clearly see faster convergence of the finite-dimensional method compared to the infinite-dimensional one for NFE = 2 and 4, and the figure only reflects mid-training behavior; showing full training curves would help clarify whether the claimed convergence advantage persists up to convergence.
>
> **Answer:**
>
> We stated the faster convergence observation of the finite-dimensional case specifically for NFE=1, as the infinite-dimensional ($D=\infty$) FID curve lies above its finite-dimensional counterparts ($D=128$ and $D=2048$). Although these curves do not reach a plateau, the behavior during the start and mid-training phases is interesting.
>
> In our work, we hypothesize that this occurs due to the finite-dimensional model's better ability to handle situations where the generator's distribution support is far from the data distribution support. In such a case, the teacher model receives out-of-distribution (OOD) samples as input and, as a result, produces poor training feedback. According to our hypothesis, finite-$D$ advantage stems from the finite-dimensional teacher's greater robustness to OOD samples, as stated in the PFGM++ work.
>
> The fact that the difference is not as noticeable for NFE=2 or 4 could be explained by the generator has more capacity and overcomes the critical distribution mismatch faster.

---

### Official Review · Reviewer_iL32 · 2025-10-30

**Soundness:** 3
**Presentation:** 3
**Contribution:** 2
**Rating:** 4
**Confidence:** 3

**Summary:**

The paper proposes Inverse Poisson Flow Matching, a distillation framework for electrostatic generative models, specifically PFGM++, to reduce the number of function evaluations needed at sampling time. The key idea is to view distillation as an inverse problem: given a teacher electrostatic field (learned on real data via PFGM++), learn a generator whose induced electrostatic field matches the teacher’s field. The authors show that the original constrained problem is equivalent to a tractable minimax objective and further express it in a denoising form. They also prove that, in the diffusion limit $D \to \infty$, their objective is closely related to Score Identity Distillation (SiD), which "justifies" reusing regularization style and hyperparameters.

**Strengths:**

- The paper gives a clear and principled formulation of distilling electrostatic models as an inverse flow matching problem.
- The connection to SiD in the $D \to \infty$ limit is well motivated.
- In the experiments presented, the method shows clear acceleration.

**Weaknesses:**

- Despite the elegant reformulation, the final training problem is still a minimax / alternating optimization over the generator and student denoiser. To me, this seems unstable and (as acknowledged by the authors) much more expensive than training the teacher model.
- The claim that one can directly transfer SiD hyperparameters to the general (finite-D) IPFM setting is only a heuristic rather than a theorem. The paper itself reports architecture-dependent divergence for $\alpha=1.2$, which doesn't really support the universality of this transfer.
- The role of the weighting function is underexplored. Authors just inherit EDM/SiD choices and reparameterize via $r = \sigma \sqrt D$. There’s no ablation to show robustness to \lambda, nor results for other data types or samplers.
- The experimental section is narrow: two unconditional image datasets, no higher-resolution or conditional generation, and no direct comparison to other fast distillation methods on the same teacher.
- IPFM applies only to electrostatic generative models, which remain niche compared to mainstream diffusion or flow models—so its practical impact is currently limited.

**Questions:**

1. Theorem 3.2 turns the constrained problem into a minimax one, but in practice, this still means alternating updates to the student and generator. Do the authors observe GAN-like instabilities (cycling, student overpowering the generator, or vice versa), and do they need multiple student steps per generator step? Some clarification on the actual training schedule would help assess feasibility.

2. The paper argues that the D \to \infty case “recovers” SiD and thus hyperparameters can be transferred. But for finite D, the perturbation kernel is heavier-tailed and the geometry is different. Do the authors have evidence that hyperparameters from SiD fail only because of architecture mismatch (ncsn++ vs ddpm++), or are there D-specific failure modes too?

3. How sensitive is IPFM to the choice of $\lambda(r)$? In particular, if the generator early in training produces very off-manifold samples, does the EDM-style $\lambda$ still give a stable learning signal to the student, or would a schedule that upweights large $r$ be preferable?

4. Are there ways to freeze the student for several iterations or to use something like the EMA students to reduce the 3x memory/cost burden?

Minor:
- The related work section is over two pages out of nine. Although quite informative, it would be more valuable to have more experiments/important discussions about the method instead.
- Please, in the main text after Theorem 3.2, refer the reader to its proof in the appendix.

I believe that these questions are important to answer and also that, if the authors (i) add a subsection on $\lambda$ choices and sensitivity, (ii) include at least one head-to-head with SiD (same dataset, same budget), and (iii) slightly expand the experiments to more datasets or a conditional setting, the paper could be stronger.

---

> ### Author Response · Authors · 2025-11-20
> **Answer to Reviewer iL32**
>
> **Questions 2 + Weakness 2:** The paper argues that the $D \to \infty$ case “recovers” SiD and thus hyperparameters can be transferred. But for finite D, the perturbation kernel is heavier-tailed and the geometry is different. Do the authors have evidence that hyperparameters from SiD fail only because of architecture mismatch (ncsn++ vs ddpm++), or are there D-specific failure modes too?
>
> **Answer:**
>
> **Hyperparameters adaptation.** We did not encounter any instability or failure related to hyperparameter selection. We reused or adapted all hyperparameters tuned for SiD (see Section 3.4 and Appendix B in our paper). This is justified for two reasons:
>
> 1. *Connection to SiD at $D \to \infty$.*
> Our IPFM recovers Score Identity Distillation (SiD) in the diffusion limit $D \to \infty$. Therefore, the hyperparameters established for SiD can be reused directly in this regime, as they were carefully tuned and validated by its authors.
>
> 2. *Valid hyperparameters transfer to finite $D$ through PFGM++ reparameterization.*
> Following PFGM++, we adopt the mapping $r = \sigma \sqrt{D}$ to transfer hyperparameters from the diffusion case to finite-$D$ settings (see Appendix B in our paper or Section 4 in [1]). This reparameterization aligns the phase [2] structure of the perturbation kernels: although finite-$D$ kernels are heavier-tailed, their effective phase progression, i.e., how much information each noise level retains about the clean data, remains consistent with the Gaussian case. Consequently, a diffusion schedule $\sigma_i$ translates naturally to a PFGM++ schedule $r_i = \sigma_i \sqrt{D}$, maintaining coherent behavior across different values of $D$.
>
> **ncsn++ vs ddpm++**. Regarding the divergence observed for the ncsn++ architecture when $\alpha=1.2$, we attribute it to architectural rather than geometric factors. This conclusion is supported by two observations:
>
> 1. Divergence occurs for both finite and infinite-$D$ (SiD) settings when using ncsn++.
>
> 2. Divergence does not occur at all when using ddpm++.
>
> Therefore, we conclude that the divergence stems from architectural limitations, not from $D$-specific failure modes or differences in the perturbation kernel’s geometry.
>
> ---
>
> **Questions 3 + Weakness 3 + Comment 1:** How sensitive is IPFM to the choice of $\lambda(r)$? In particular, if the generator early in training produces very off-manifold samples, does the EDM-style $\lambda$ still gives a stable learning signal to the student, or would a schedule that upweights large $r$ be preferable?
>
> **Answer:**
>
> As we previously mentioned, we did not encounter issues with hyperparameter selection. We used well-tuned SiD hyperparameters and adapted them to finite-$D$ via the $r = \sigma \sqrt{D}$ parameterization.
>
> However, we agree that early training can be sensitive, since off-manifold generator samples lead the teacher to process out-of-distribution (OOD) inputs. The weighting function $\lambda(r)$ (defined as $\lambda(r) = \lambda_\sigma(r/\sqrt{D})$) and sampling schedule $p(r)$ (defined via $p_\sigma$ as discussed in Appendix B of our paper) play important roles here ($\lambda_\sigma$ and $p_\sigma$ are hyperparameters for the diffusion $D\to\infty$ setting). There are two places where $\lambda_\sigma(\sigma)$ and $p_\sigma(\sigma)$ matter:
>
> 1. *Student flow training.*
> Here, the student simply learns the normalized field induced by the generator distribution. Its stability does not depend strongly on how far the generator is from the data manifold. Following SiD [4], we reuse the teacher's $\lambda_\sigma(\sigma)$ and $p_\sigma(\sigma)$ for training the student flow.
>
> 2. *Generator training.*
> Here, the discrepancy between generator and data distributions is more critical. When using only low noise level $\sigma$ values (where noise does not sufficiently reduce OOD effects), the teacher produces a poor training signal that prevents convergence. Conversely, using only high noise level $\sigma$ values (where little information about the original distribution remains in the noised samples) reduces OOD issues but oversmooths fine details.
>
> Recent works [4,5] (including ours) use an adaptive weighting (Eq. (37)) that downweights samples where the teacher reconstructs poorly: either due to high noise or OOD inputs. This stabilizes training without explicitly upweighting large $r$.
>
> We use the SiD-style fixed noise schedule (Eq. (36)), which focuses more on low noise levels and can slow early convergence slightly due to OOD samples. However, in Section 4 we observe that finite-$D$ models converge faster than their $D \to \infty$ counterpart (the diffusion distillation method SiD). We attribute this to the greater stability of lower-$D$ models, which is inherited from the heavier tails of their perturbation kernels.

---

> ### Author Response · Authors · 2025-11-20
> **Answer to Reviewer iL32**
>
> **Questions 1 + Weakness 1:** Does the minimax formulation of IPFM lead to GAN-like training instabilities, and what is the practical training schedule?
>
> **Answer:**
>
> **Stability of our IPFM.** As mentioned previously, we did not encounter issues with hyperparameter selection. We used well-tuned SiD hyperparameters and adapted them to finite-$D$ via the $r = \sigma \sqrt{D}$ parameterization. **In Figure 2, we plot FID versus training step, which demonstrates largely monotonic convergence.**
>
> While the minimax structure resembles GAN training, we did not observe GAN-like instabilities. We hypothesize this is due to the following differences from GANs:
>
> 1. Our IPFM generator is initialized with weights from pre-trained teacher models, rather than random initialization as in GANs.
>
> 2. The student flow learns the normalized field induced by the generator's distribution, rather than discriminating between real and fake samples.
>
> 3. Both student and teacher flows process noisy generator samples, unlike GAN's discriminator which typically processes clean samples.
>
> **Update schedule.** Regarding the update schedule, we follow SiD and use one student update per generator update, which we found sufficient. With good initialization from the teacher and relatively slow generator updates, the student accurately tracks the normalized flow induced by the current generator. Although performing multiple student steps per generator step would be more faithful to the underlying optimization structure, we did not find it necessary in practice.
>
> ---
>
> **Questions 4:** Are there ways to freeze the student for several iterations or to use something like the EMA students to reduce the 3x memory/cost burden?
>
> **Answer:**
>
> Improving the efficiency of methods such as IPFM and SiD is indeed an important research direction. Techniques that reduce the memory or computational cost for SiD would likely be applicable to IPFM in the finite-$D$ case as well. However, to the best of our knowledge, no prior work has yet proposed such modifications.
>
> *Regarding the specific suggestions.* Simply freezing the student for several iterations or using an EMA student may adversely affect performance. Both techniques would slow the student's updates, whereas the student update corresponds to the inner minimization and must keep pace with the generator (the outer problem) to maintain an accurate estimate of the normalized Poisson flow. A stale or slowly updated student would not provide a reliable training signal to the generator.
>
> ---
>
> **Weakness 4 + Comment 3:** Narrow experimental section, no direct comparison to other fast distillation methods on the same teacher.
>
> **Answer:**
>
> **Benchmark.** We agree that a broader experimental evaluation (e.g., higher-resolution and more conditional settings) would further strengthen the paper. Our current setup is mainly constrained by computational resources. In the revised version, we additionally report IPFM results for the class-conditional CIFAR-10 PFGM++ teacher, thus **covering all teacher models released by the PFGM++ authors**. The new class-conditional results are consistent with our unconditional findings. **We have updated the revised manuscript** with this results in the Appendix D.
>
> **Baselines.** To the best of our knowledge, SiD is the only GAN-free distillation method that can match (or even surpass) the performance of a diffusion teacher (the $D \to \infty$ regime of our IPFM). Our results show that finite-$D$ variants of our IPFM converge at least as well as, and in some cases better than, this diffusion limit. This indicates that our IPFM can attain teacher-level performance in a single step given sufficient compute.  By contrast, other GAN-free distillation approaches (e.g., consistency-based models) have not been demonstrated to reach teacher performance even for diffusion teachers. Moreover, there are no specific adaptations of these approaches for PFGM++ teachers on standard generative benchmarks. For this reason, we do not include other baselines for finite-$D$ Poisson-flow distillation and focus our evaluation on the theoretically grounded connection between our IPFM and SiD across different values of $D$. **We have updated the revised manuscript** with this discussion in the Discussion section.

---

> > ### Author Response · Authors · 2025-11-20
> > **Answer to Reviewer iL32**
> >
> > **Comment 2:** Include at least one head-to-head with SiD (same dataset, same budget).
> >
> > **Answer:**
> >
> > We would like to clarify that our current experiments already contain a head-to-head comparison with SiD. An IPFM configuration with $D \to \infty$ and $\alpha = 1.0$ reduces exactly to SiD. In practice, this means that the $D \to \infty$ IPFM setting coincides with SiD under the same dataset, architecture, NFE budget, and hyperparameters. Across the rows in our tables, we keep these choices and the computational budget aligned; the only differences are the teacher (finite-$D$ PFGM++ vs. diffusion) and the corresponding auxiliary dimension $D$. As shown in Figure 2, our IPFM with a finite-$D$ one-step generator converges faster than the one-step SiD.
> >
> > ---
> >
> > **Comment 4:** Please, in the main text after Theorem 3.2, refer the reader to its proof in the appendix.
> >
> > **Answer:**
> >
> > Please note that at the beginning of the section we included a reference to the section containing all proofs. If you believe this is not sufficiently convenient, please let us know, and we will gladly add a reference right after Theorem 3.2.
> >
> > ---
> >
> > **Conclusion remarks:**
> >
> > We hope these clarifications have fully addressed your concerns. Should any points require further elaboration, we are happy to provide it. Based on these comprehensive explanations, we are hopeful that you will consider raising your rating.
> >
> > ---
> >
> > [1] PFGM++: Unlocking the Potential of Physics-Inspired Generative Models
> > [2] Stable target field for reduced variance score estimation in diffusion models
> > [3] Elucidating the Design Space of Diffusion-Based Generative Models
> > [4] Score identity Distillation: Exponentially Fast Distillation of Pretrained Diffusion Models for One-Step Generation
> > [5] One-step Diffusion with Distribution Matching Distillation

---

### Official Review · Reviewer_9142 · 2025-10-31

**Soundness:** 4
**Presentation:** 1
**Contribution:** 3
**Rating:** 4
**Confidence:** 4

**Summary:**

The paper introduces Inverse Poisson Flow Matching (IPFM), a novel method derived from first principles for distilling Poisson Flow Generative Models (PFGM++). The authors establish a theoretical connection between IPFM and Score Identity Distillation (SiD), a technique used for diffusion models. Leveraging this connection, they successfully adapt a regularizer from SiD, which is shown to empirically benefit IPFM.

Empirically, the distilled models achieve performance competitive with the original teacher model, and in some cases, even appear to surpass it. A significant advantage of the proposed method is that it is data-free, requiring no access to the original training dataset for the distillation process.

**Strengths:**

While model distillation is a well-studied field for standard diffusion models, its application to Poisson Flow Generative Models (PFGM++) remains largely unexplored. The primary strength of this paper is its rigorous, first-principles derivation of a distillation method specifically tailored for PFGM++.

Instead of merely adapting existing diffusion model techniques, the authors provide a principled validation for their approach. This theoretical grounding builds confidence in the method's correctness and suitability for the PFGM++ framework. Although PFGM++ is a more specialized approach compared to mainstream diffusion models, this work represents a significant contribution by addressing an important and under-researched area in generative model distillation.

**Weaknesses:**

The paper's primary weakness is the omission of a crucial and highly relevant prior work, [1], which adapts Consistency Distillation for PFGM++. While the authors do not explicitly claim to be the first to distill PFGM++, the manuscript is framed in a way that implies this, and the lack of discussion or comparison to [1] is a significant oversight.

For a comprehensive evaluation, an empirical comparison against the method proposed in [1] is essential. At a minimum, the authors must contextualize their work by thoroughly discussing this existing approach. Without this comparison and discussion, it is difficult to assess the relative contribution and performance of IPFM, which hinders a positive evaluation of the submission.

Minor Comments:
- The empirical validation would be strengthened by including experiments on larger-scale datasets to further assess the method's scalability and robustness.
- The repetition of Equation (21) as Equation (22) with minimal separation appears redundant and could be consolidated.

[1] Hein, Dennis, et al. "PFCM: Poisson flow consistency models for low-dose CT image denoising." IEEE Transactions on Medical Imaging (2025).

**Questions:**

1. The proposed method is named "Inverse Poisson Flow Matching" (IPFM). Why did the authors choose not to feature this name in the paper's title, given that it is the central contribution?
2. Regarding the notation on Line 99: should the term x′ be x (without the prime)? Please clarify this potential typo.
3. In Table 1, the distilled models are compared to the teacher model at 35 NFE (CIFAR-10) and 79 NFE (FFHQ). The authors claim "even superior sample quality" for the distilled model. However, it is unclear if the teacher model's performance has saturated at this NFE. Could the authors provide results for the teacher model at a higher NFE to confirm that this is a fair comparison and that the distilled model is not just outperforming a sub-optimal, non-converged teacher?

---

> ### Author Response · Authors · 2025-11-20
> **Answer to Reviewer 9142**
>
> **Question 1:** The proposed method is named "Inverse Poisson Flow Matching" (IPFM). Why did the authors choose not to feature this name in the paper's title, given that it is the central contribution?
>
> **Answer:**
>
> You are absolutely right that featuring Inverse Poisson Flow Matching (IPFM) in the title would be a natural choice. We considered such options, but ultimately aimed for a title that would be clearer and more accessible to a broader audience. Since Poisson Flow–based generative models are still relatively unfamiliar to many readers, we felt that “Overclocking Electrostatic Generative Models” better conveys the paper’s focus while remaining concise and intuitive.
>
> ---
>
> **Question 2:** Regarding the notation on Line 99: should the term x′ be x (without the prime)?
>
> **Answer:**
>
> Thank you for the question. There is no typo. In Eq. (1), we follow notation from [3]:
>
> - $\mathbf{x}' \in \mathbb{R}^N$ denotes the location of the source point charge,
> - $\mathbf{x} \in \mathbb{R}^N$ denotes the evaluation point where the electric field is computed.
>
> A point charge at $\mathbf{x}'$ induces an electric field $\mathbf{E}(\mathbf{x})$ at every point $\mathbf{x}$, and this field is given by Eq. (1). Thus, the prime notation is intentional and distinguishes the source position from the evaluation position.
>
> For better clarity and consistency with the subsequent parts of the paper, **we have updated the revised manuscript** by moving the source point charge sign from $\mathbf{x}'$ to $\mathbf{y}$.
>
> ---
>
> **Question 3:** Evaluation of teacher models with increased NFE.
>
> **Answer:**
>
> This is a fair concern. We initially adopted the NFE values from the original works [1, 2].
>
> To directly address your question, we evaluated the teacher models at higher NFE values (99, 199, and 999). The tables below show that the teacher FID at higher NFEs (even 999) does not significantly differ from the original values, indicating that the teacher performance is already saturated around the NFEs used in Table 1.
>
> **We have included these results in Appendix D of the revised manuscript.**
>
> **CIFAR-10 (FID)**
>
> | Dataset     |  D   | Our Best IPFM | Teacher (NFE = 35) | Teacher (NFE = 99) | Teacher (NFE = 199) | Teacher (NFE = 999) |
> |--|:--:|:--:|:--:|:--:|:--:|:--:|
> | CIFAR-10   | 128  | **1.75**         | 1.92*/1.92         | 1.94               | 1.97                | 1.97                |
> | CIFAR-10   | 2048 | **1.82**         | 1.91*/1.91         | 1.95               | 1.95                | 1.97                |
> | CIFAR-10   | $\infty$ | **1.86**     | 1.98*/1.99         | 1.99               | 2.01                | 2.02                |
>
> **FFHQ (FID)**
>
> | Dataset |   D    | Our Best IPFM | Teacher (NFE = 79) | Teacher (NFE = 99) | Teacher (NFE = 199) | Teacher (NFE = 999) |
> |--|:--:|:--:|:--:|:--:|:--:|:--:|
> | FFHQ    | 128    | **1.72**         | 2.43*/2.47         | 2.46               | 2.45                | 2.44                |
> | FFHQ    | $\infty$ | **1.70**       | 2.53*/2.53         | 2.53               | 2.52                | 2.52                |
>
> *Note:* For the original NFEs (35 for CIFAR-10 and 79 for FFHQ), we report two values in the teacher columns. The value marked with `*` is taken directly from the original paper and was computed according to [1, 2] (as described in Appendix B of our paper). The second value corresponds to computing FID a single time on 50,000 generated images, consistent with how we evaluated the other teacher NFEs.
>
> ---
>
> **Weakness:** Missed Poisson Flow Consistency Models as relevant related work.
>
> **Answer:**
>
> Thank you for pointing this out. We agree that PFCM is a related work and deserves discussion in the paper. PFCM also provides supporting evidence that finite-$D$ electrostatic models can be advantageous, which is consistent with our findings.
> **We have updated the revised manuscript** in the "Introduction" part to highlight this.
>
> However, a direct empirical comparison is not straightforward due to the different tasks and evaluation settings. PFCM is a more application-focused work, as it is designed and evaluated for paired CT image denoising, a supervised image-to-image task on medical data. In contrast, our work is more methodological, as we primarily focus on standard benchmarks for general generative modeling.
>
> **We have updated the manuscript** with a discussion comparing our method to other baselines in the Discussion section.
>
> ---
>
> **Conclusion remarks:**
>
> We hope these clarifications have fully addressed your concerns. Should any points require further elaboration, we are happy to provide it. Based on these comprehensive explanations, we are hopeful that you will consider raising your rating.
>
> ---
>
> [1] PFGM++: Unlocking the Potential of Physics-Inspired Generative Models
> [2] Elucidating the Design Space of Diffusion-Based Generative Models
> [3] Field Matching: an Electrostatic Paradigm to Generate and Transfer Data

---

> > ### Comment · Reviewer_9142 · 2025-11-28
> >
> > Dear Authors,
> >
> > Thank you for your detailed response and for updating the manuscript. I decided to raise my rating by one level.
> >
> > Regarding PFCM: I'm raising my rating because this method was referenced in the updated manuscript. I'd be willing to raise it even higher if an empirical comparison of the methods were available. I understand that this would require additional work and experimentation by the authors. However, the fact that the competing method was presented on different datasets does not change the fact that, as a reviewer and reader, I expect a detailed comparison.
> >
> > Kind regards,

---

> > > ### Author Response · Authors · 2025-12-03
> > > **Answer to Reviewer 9142**
> > >
> > > First of all, we would like to thank you for increasing the rating.
> > >
> > > Consering comparison between general ODE-based distillation and our IPFM, we agree that more explicit comparision would better emphasize the superiority of our IPFM. To this end, we added a sanity-check experiment comparing our IPFM to Consistency Trajectory Models, an advanced consistency-based distillation, in finite-$D$ setting. The results show the superior performance of our IPFM, highlighting the importance of our method. **We have updated the revised manuscript** with this experiment in the Appendix D and added relevant comments in the Introduction and Discussion sections.

---

### Official Review · Reviewer_7SD5 · 2025-11-01

**Soundness:** 3
**Presentation:** 1
**Contribution:** 3
**Rating:** 4
**Confidence:** 3

**Summary:**

The authors proposed  a distillation method for PFGM++ a model which is somewhat tangential to diffusion models (and is identical in the limit of $D \to \infty$). Like in prior work the authors observe that small values of $D$ result in faster convergence for the distilled models.

**Strengths:**

* Explores distillation for a novel class of generative models (PFGM and PFGM++)
* Non-trival results
* Promising results on faster convergence with small $D$
* Highly interesting topic

**Weaknesses:**

## Primary Concerns
I like this paper and would be inclined to give it a higher score, if the explanation of the core concepts are more intuitively illustrated.

It is very hard to visualize what PGFM++ does for generative modeling and consequently what the contributions are. What role does $\mathbf z \in \mathbb R^D$ play, why is it necessary. What is the physically meaning of $\mathbf z$? My rough understanding is that as $D \to \infty$, $r \to \sigma$ or $t$?

Most of my questions regarding clarity reside with the questions section so I will refer the authors to that section.

Really my only primary concern is the clarity in communicating the concepts well so the reader can understand it.

**Questions:**

1. It isn't clear to me that in line 150 $f_\phi: R^{N+1} \to \mathbb R^{N+1}$. Wouldn't it be $f_\phi: R^{N+D} \to \mathbb R^{N+D}$?
2. How do you pick $r_max$? What is it physical meaning.
3. The claim that PFGM++ is diffusion in the limit of $D \to \infty$ is not obvious, why is this? What is the physical meaning?
4. What does integrating along $r$ mean?
5. In Figure 1 and Section 3.1 the authors state the goal is to learn $G_\theta$, but I don't see it in any of the equations in Section 3.1 or 3.2. Where is it and what does it do? From Figure 1 I suppose the signature is $G_\theta: \mathbb R^D \to \mathbb R^{N+D}$ but this is never explicitly stated. What is it?
6. Isn't the goal to learn $f_\psi$ not $G_\theta$ if this understanding is wrong, why? **It is not clear from the text.**
7. In Table 1 the authors state that $D = \infty$, but this doesn't seem tractable. Again clarity.

---

> ### Author Response · Authors · 2025-11-20
> **Answer to Reviewer 7SD5**
>
> Thank you for these questions. Electrostatic-based generative models are relatively new and not yet widely known, so it is important to introduce the main concepts clearly while remaining within the space constraints. Generally, while describing the PFGM++ framework we follow the original paper's notation. Thus, along with providing answers to your questions, we leave references to the original paper's sections.
>
> ---
>
> **Question 1:** It isn't clear to me that $f_\phi: \mathbb{R}^{N + 1} \to \mathbb{R}^{N + 1}$. Wouldn't it be $f_\phi: \mathbb{R}^{N + D} \to \mathbb{R}^{N + D}$?
>
> **Answer:**
>
> *More details can be found in [1, Section 2] or in our Section 2.*
>
> In the electrostatic construction, the data are embedded in the augmented space $\mathbb{R}^{N + D}$ with coordinates $(x, z)$, where $x \in \mathbb{R}^N$ are data coordinates and $z \in \mathbb{R}^D$ are auxiliary dimensions. Because the charge configuration is rotationally symmetric in the $z$–subspace, the electrostatic field depends on $z$ only through its radius $r = \lVert z \rVert_2.$
>
> Therefore, all quantities relevant for sampling can be expressed in the reduced coordinates $\tilde x = (x, r) \in \mathbb{R}^{N+1},$ and the field-line ODE (Eq. (3)) can be written as $\frac{dx}{dr} = \frac{E_x(\tilde x)}{E_r(\tilde x)}$ (Eq. (5)).
>
> The network $f_\phi$ is trained to approximate the (normalized) Poisson field in this reduced $\mathbb{R}^{N+1}$ space, so its domain and codomain are $\mathbb{R}^{N+1}$, not $\mathbb{R}^{N+D}$. Note that this reduction changes only the parameterization of the inputs: the dependence on $D$ is still fully present in the electrostatic field and perturbation kernel (Eq. (10)), and thus implicitly in the target that $f_\phi$ is trained to match.
>
> **We have updated the revised manuscript** at the end of the "Augmented space" paragraph in Section 2 to highlight that PFGM++ models work with objects in $\mathbb{R}^{N+1}$ and not $\mathbb{R}^{N+D}$.
>
> ---
>
> **Question 4:** What does integrating along $r$ mean?
>
> **Answer:**
>
> *More details can be found in [1, Section 3.1] or in our Section 2.*
>
> In our setting, the data lie on the hyperplane $\{z = 0\}$ in the augmented space $\mathbb{R}^{N+D}$, and $r = \lVert z \rVert_2$ is the radial distance from this hyperplane. Given this setup, prior distribution $p_{r_\max}$ lies in the space at distance $r_\max$ from the data hyperplane.
>
> We solve the ODE from $r = r_{\max}$ (where the prior is defined) down to $r = 0$ (the data hyperplane), as discussed around Eqs. (5)–(7) in Section 2.2. In this sense, $r$ plays a role analogous to the time/noise level $t$ or $\sigma$ in diffusion models.
>
> ---
>
> **Question 3:** The claim that PFGM++ is diffusion in the limit of $D \to \infty$ is not obvious, why is this? What is the physical meaning?
>
> **Answer:**
>
> *More details can be found in [1, Section 4] or in our Section 2.*
>
> This limit is easiest to see from the PFGM++ perturbation kernel (Eq. (10)). For fixed $r$, the kernel has polynomial tails in $\lVert x - y\rVert_2^2$, with the decay exponent growing with $D$. If we reparameterize $r$ as $r = \sigma \sqrt{D}$ and let $D \to \infty$, one can show that the perturbation kernel converges to the Gaussian kernel of a diffusion models (Eq. (15), which has exponential tails in $\lVert x - y\rVert_2^2$): adding noise by sampling from the PFGM++ kernel becomes equivalent to adding Gaussian noise with variance $\sigma^2$.
>
> As proved in PFGM++ paper, in this limit, not only the kernel but also the sampling ODE and the training objective converge to those of a standard diffusion model (Eqs. (14) and (16)), so PFGM++ becomes exactly a diffusion model with noise level $\sigma$.

---

> ### Author Response · Authors · 2025-11-20
> **Answer to Reviewer 7SD5**
>
> **Question 2:** How do you pick $r_{\max}$? What is its physical meaning?
>
> **Answer:**
>
> *More details can be found in [1, Section 4] or in our Section 2 and Appendix B.*
>
> As discussed previously, the prior distribution $p_{r_\max}$ lies in the space at distance $r_\max$ from the data hyperplane. It plays a role similar to $\sigma_{\max}$ in diffusion models: it is the maximum noise level (or distance) from which sampling trajectories start before flowing back to $r = 0$ (the data hyperplane).
>
> To choose $r_{\max}$ for a given $D$, PFGM++ proposes transferring well-tuned diffusion hyperparameters (the $D \to \infty$ case) to finite $D$ via the reparameterization $r = \sigma \sqrt{D}$. This aligns the “phase structure” [2] of the perturbation kernels: although finite-$D$ kernels are heavier-tailed, the amount of information retained about the clean data at each noise level remains consistent with the Gaussian case.
>
> Concretely, following PFGM++, we set $r_{\max} = \sigma_{\max} \sqrt{D}$, where $\sigma_{\max}$ is taken from the diffusion baseline [3].
>
> ---
>
> **Question 5:** Why is the generator $G_\theta$ not explicitly present in the IPFM problem statement?
>
> **Answer:**
>
> We use generator $G_\theta$ in the standard way to define a parametrized distribution $p_\theta$ from which we only need to draw samples. Concretely, we sample a simple prior noise $\epsilon \sim p_{\text{prior}}$ and set $y = G_\theta(\epsilon).$ This induces a distribution $p_\theta(y)$: $y \sim p_\theta(y),$ which is exactly the pushforward of $p_{\text{prior}}$ through $G_\theta$.
>
> In Sections 3.1–3.2, $G_\theta$ appears implicitly through $p_\theta(y)$: all expectations of the form $\mathbb{E}\_{y \sim p_\theta(y), r, x_r}[\cdot]$ depend on $\theta$ via this pushforward, even though $G_\theta$ is not written explicitly inside the equations.
>
> **We have updated the revised manuscript** within the Inverse Poisson Flow Matching problem definition to resolve this ambiguity.
>
> ---
>
> **Question 6:** Isn't the goal to learn $f_\psi$ not $G_\theta$?
>
> **Answer:**
>
> During the distillation process, we simultaneously train two models:
> 1) The few-step generator $G_\theta$
> 2) The Poisson flow $f_\psi$ induced by the generator distribution $p_\theta$
>
> The final goal is indeed to learn the few-step generator $G_\theta$, not $f_\psi$. This is evident from the Inverse Poisson Flow Matching objective (Eq. (19)), where the outer optimization is over the generator parameters $\theta$.
>
> Although both models can generate samples from $p_\theta$, we ultimately require only the generator $G_\theta$ because it can produce samples **in few steps**. In contrast, $f_\psi$ requires simulating the full ODE (Eq. (5)). Moreover, we already have a teacher model $f^*$ for that purpose. Therefore, $f_\psi$ in this objective serves as an auxiliary model.
>
> We believe the **manuscript update** addressing Question 5 also clarifies this point.
>
> ---
>
> **Question 7:** In Table 1 the authors state that $D = \infty$, but this doesn't seem tractable. What does it mean?
>
> **Answer:**
>
> We follow the PFGM++ notation [1], where $D \to \infty$ represents the theoretical limit that recovers standard diffusion models [3]. In practice, we do not substitute $D = \infty$ into PFGM++ formulas. Instead, we directly use the diffusion model components that emerge in this limit: the Gaussian perturbation kernel, diffusion training objective, and corresponding sampling ODE.
>
> In other words, the $D \to \infty$ entry in Table 1 refers to using established diffusion methodology, which represents the limiting case of the PFGM++ framework.
>
> ---
>
> **Conclusion remarks:**
>
> We hope these clarifications have fully addressed your concerns. Should any points require further elaboration, we are happy to provide it. Based on these comprehensive explanations, we are hopeful that you will consider raising your rating.
>
>
> ---
>
> [1] PFGM++: Unlocking the Potential of Physics-Inspired Generative Models
> [2] Stable target field for reduced variance score estimation in diffusion models
> [3] Elucidating the Design Space of Diffusion-Based Generative Models

---

> > ### Comment · Reviewer_7SD5 · 2025-11-23
> >
> > The authors have made great efforts to improve their clarity and I will update my score accordingly.

---

### Author Response · Authors · 2025-11-20
**Revised paper**

We sincerely thank all reviewers for their constructive and insightful feedback. We have uploaded an updated version of the paper; all changes are highlighted in **blue**. The main changes are:

[**iL32**, **1SzS**] We added **evaluation of our IPFM on class-conditional CIFAR-10**. The obtained results are consistent with those for unconditional generative tasks. Quantitative results can be found in Appendix D, and qualitative results are provided in Appendix E.

[**1SzS**] We added a **theorem showing that any minimizer of our IPFM problem must coincide with the data distribution**. The theorem is stated right after the IPFM problem definition, and its proof is placed in Appendix A.

[**9142**, **iL32**, **1SzS**] We **expanded the discussion comparing our IPFM with other baselines** in the “Discussion” section. We also **added references to the related Poisson Flow Consistency Models**, which adapt Consistency Distillation to PFGM++ for low-dose CT image denoising and show strong performance in finite-$D$ settings—consistent with our observations.

[**iL32**, **1SzS**] We added two additional experiments: (1) **a regularization-strength $\alpha$ ablation for the finite-$D$ case**, showing trends consistent with the SiD observations, and (2) **evaluation of the teacher models with higher NFE values**, confirming that the results reported in the main text already reflect their saturated performance. These experiments are presented in Appendix D.

[**7SD5**, **9142**, **iL32**, **1SzS**] We added further clarifications to the main text to address the points raised in the reviews.

Please consider the updated revision. We are happy to address any remaining points during the discussion phase.

---

### Author Response · Authors · 2025-12-03
**Revised paper**

We thank the reviewers for their valuable suggestions.
We have uploaded an updated version of the paper; all changes for this revision are highlighted in **orange**. The main updates are:

[**9142**,**1SzS**] We added a comparison between Consistency Trajectory Models and our IPFM in a finite-$D$ setting. The results demonstrate the superior performance of our IPFM. Details are provided in Appendix D.

[**1SzS**] We revised the proof of Theorem 3.2 to improve its structure and clarity, as suggested. The updated proof is presented in Appendix A.

We believe we have addressed all concerns raised in the reviews.

---

We note that the main changes in the revised version of our paper are listed in the General Responses. Detailed discussion of these changes can be found in our responses to the corresponding reviewers.

Since one of the main comments from the reviewers regarded limited benchmarks, we would like to note that we have addressed these concerns in our revised version; we have now distilled all PFGM++ teacher models provided by the authors.

---

### Meta-Review · Area_Chair_iwLD · 2026-01-08

**Summary:**

This paper proposes Inverse Poisson Flow Matching (IPFM), a distillation framework for accelerating sampling in electrostatic generative models (PFGM++). By formulating distillation as an inverse problem, i.e. matching the electrostatic field induced by the generator, the authors derive a tractable objective that recovers Score Identity Distillation (SiD) in the diffusion limit.

While the initial reviews were lukewarm (three 4s, one 6), with the consensus acknowledging the theoretical novelty of the inverse formulation and the solid connection to SiD. Major concerns centered on the lack of comparisons to trajectory-based distillation baselines (e.g., Consistency Models), missing proofs for the objective's well-posedness, narrow experimental scope (limited datasets/teachers), and poor presentation clarity regarding the core concepts of PFGM++.

**Reviewer Concerns:**

Addressed:
- Theoretical Well-Posedness (1SzS): The authors added a proof in Appendix A. However, the reviewer noted the proof used informal notation and was "hard to follow," limiting the credit given for this fix.
- Clarity (7SD5): Basic notations were clarified, satisfying this specific reviewer.

Partially Addressed:
- Baselines (9142, 1SzS): The authors added a comparison to CTM in the revision. However, Reviewer 9142 explicitly noted they would have rated the paper higher only if a direct comparison to the most relevant prior work (PFCM) was included, which the authors argued was out of scope due to datasets. This gap prevented a stronger score increase.
- Convergence Claims (1SzS): The reviewer challenged the claim that finite-$D$ models converge faster, stating the plots didn't show it clearly. The authors offered a hypothesis but didn't provide the "full training curves" requested to empirically prove it, leaving the claim unsubstantiated.

Outstanding:
- Impact/Scope (iL32): The method is specific to electrostatic models. Since these are not the community standard, a distillation method for them has limited utility compared to universal distillation techniques
- Proof Quality (1SzS): The reviewer remained critical of the "unpolished" nature of the new theoretical content even after the rebuttal.

**Reviewer Scores:**

- Reviewer 7SD5 (Original: 4): Predicted: 6 (Marginally Above). Raised score based on clarity fixes.

- Reviewer 9142 (Original: 4): Predicted: 6 (Marginally Above). Raised score for the citation of PFCM, but withheld a higher score due to the lack of empirical comparison to it.

- Reviewer iL32 (Original: 4): Predicted: 4 (Marginally Below). The reviewer's concern about the niche scope and lack of broad benchmarks was not fundamentally resolved.

- Reviewer 1SzS (Original: 6): Predicted: 6 (Marginally Above). Explicitly stated: "I prefer to keep my original score" despite the revisions, citing unpolished proofs and unconvincing convergence plots.

---

### Decision · Program_Chairs · 2026-01-26

Reject